# The signaling axis atypical protein kinase C λ/ι-Satb2 mediates leukemic transformation of B-cell progenitors

R.C. Nayak[1], S. Hegde [1,2], M.J. Althoff[1,2,3], A.M. Wellendorf[1], F. Mohmoud[3], J. Perentesis[4], M. Reina-Campos[5], D. Reynaud[1], Y. Zheng[1], M.T. Diaz-Meco [5], J. Moscat [5] & J.A. Cancelas[1,2,3]

Epigenetically regulated transcriptional plasticity has been proposed as a mechanism of differentiation arrest and resistance to therapy. BCR-ABL leukemias result from leukemic stem cell/progenitor transformation and represent an opportunity to identify epigenetic progress contributing to lineage leukemogenesis. Primary human and murine BCR-ABL[+] leukemic progenitors have increased activation of Cdc42 and the downstream atypical protein kinase C (aPKC). While the isoform aPKCζ behaves as a leukemic suppressor, aPKCλ/ι is critically required for oncogenic progenitor proliferation, survival, and B-cell differentiation arrest, but not for normal B-cell lineage differentiation. In vitro and in vivo B-cell transformation by BCR-ABL requires the downregulation of key genes in the B-cell differentiation program through an aPKC λ/ι-Erk dependent Etv5/Satb2 chromatin repressive signaling complex. Genetic or pharmacological targeting of aPKC impairs human oncogenic addicted leukemias. Therefore, the aPKCλ/ι-SATB2 signaling cascade is required for leukemic BCR-ABL[+] B-cell progenitor transformation and is amenable to non-tyrosine kinase inhibition.

[1] Division of Experimental Hematology, Cancer and Blood Diseases Institute, Cincinnati Children's Hospital Medical Center, 3333 Burnet Ave., Cincinnati, OH 45229, USA. [2] Hoxworth Blood Center, University of Cincinnati, 3130 Highland Ave., Cincinnati, OH 45267, USA. [3] Graduate Program of Cancer & Cell Biology, University of Cincinnati College of Medicine, Cincinnati, OH 45267, USA. [4] Division of Oncology, Cancer and Blood Diseases Institute, Cincinnati Children's Hospital Medical Center, 3333 Burnet Ave., Cincinnati, OH 45229, USA. [5] Cancer Metabolism and Signaling Networks Program, Sanford Burnham Prebys Medical Discovery Institute, 10901 North Torrey Pines Road, La Jolla, CA 92037, USA. Correspondence and requests for materials should be addressed to J.A.C. (email: jose.cancelas@uc.edu)

B lymphoid leukemia arises from hematopoietic stem cells (HSC) or B-cell progenitors, so-called leukemic progenitors that have acquired a transforming, leukemia-initiating event. A major example of a leukemia-initiating event is the expression of p210-BCR-ABL, which is the product of t(9;22) (q34;q11) translocation, and is necessary and sufficient for the development and progression of chronic myelogenous leukemia (CML)[1]. The transforming ability of BCR-ABL is dependent on its deregulated tyrosine kinase (TK) activity leading to its auto-phosphorylation, recruitment of adaptor proteins, and subsequent activation of downstream signaling pathways, including Ras, extracellular-signal-regulated kinase (ERK), Akt, c-Jun activated kinase (JNK), p38, CrkL, signal transducer and activator of transcription 5 (STAT5), and nuclear factor-κB (NF-kB)[2]. Progression of BCR-ABL[+] leukemia from the chronic phase to the poor prognosis blast crisis phase is accompanied by increased BCR-ABL expression, genetic instability, increased proliferation, reduced apoptosis, and a blockade of differentiation where myeloid or lymphoid progenitors/precursors fail to differentiate, resulting in the development of acute myelogenous leukemia (AML) or B-cell acute lymphoblastic leukemia (B-ALL)[2–5]. Genetic abnormalities such as increased Myc expression[6], upregulation of Bmi1[7], homozygous deletion of exon 2 of INK4A/ARF[8], and dominant-negative deletions and mutations of lymphoid transcription factors, including PAX5, IKZF1, and EBF1[9–11], cooperate with BCR-ABL in the differentiation arrest of lymphoid precursors.

Loss of polarity and deregulation of asymmetric cell division are hallmarks of cancer stem cells[12]. Molecularly, it has been argued that polarity complexes are involved in BCR-ABL-induced leukemogenic transformation[13,14]. In the erythroleukemia cell line K562, BCR-ABL-induced upregulation of atypical protein kinase Cι (aPKCι) expression is necessary for the resistance to drug-induced apoptosis mediated by the activation of NF-κB/RelA pathway[15,16]. However there has been a significant lack of understanding of the specific oncogenic signaling pathways used by polarity-related fate determinants in transformation and, more specifically, in oncogenic signaling-dependent lineage differentiation arrest. Atypical protein kinase Cζ (aPKCζ) and -λ/ι (aPKCλ, mouse homolog; aPKCι, human homolog) are the sole kinases in the apical polarity complex. Both aPKCs can interact with PAR6 through their respective PB1 domains, and together with PAR3 form the apical polarity complex that activate and segregate the downstream effector Numb, and dictate the asymmetric cell division of stem cells and progenitors during embryonic development and adult tissue homeostasis[12,17]. Besides regulating asymmetric cell division, the aPKCs, by binding to the adaptor protein p62 and other PB1 domain-containing proteins, regulate various signaling pathways involved in cell survival, proliferation and differentiation[18,19]. Recent analysis of somatic mutations in the different PKC isoforms led to the prediction that, in general, PKCs function as tumor suppressors[20]. In the case of PKCζ, there is abundant data to demonstrate that this is correct[21–23]. However, in the case of aPKC λ/ι, whether it acts as a tumor promoter or suppressor seems to be context-specific. For instance, aPKC λ/ι-activity is required for murine Ras induced-lung adenocarcinoma[24] but acts as a non-cell autonomous tumor suppressor in intestinal carcinogenesis[25].

Here we show that aPKCλ/ι is a critical player in the transformation of B-cell progenitors by BCR-ABL through a cascade of events that includes MEK/ERK activation, Numb translocation, and the upregulation of the chromatin modifier Satb2. Our data strongly support aPKCλ/ι as a potential therapeutic target to prevent lymphoid transformation of leukemic stem cells/progenitors (LSC/P) and inhibit TK inhibitor (TKI)-resistant CML.

## Results

**aPKCι is highly expressed and activated in LSC/P.** We first examined the expression and activation of the aPKC isoforms in primary human CML stem and progenitors (CML CD34[+]) and in B-ALL B-cell progenitors (CD34[+]/CD19[+]). We observed that while aPKCι was highly expressed, aPKCζ expression was very reduced at the mRNA (Supplementary Figure 1A) and protein (Fig. 1a, Supplementary Figure 1B) levels in CD34[+] cells from CML patients and healthy donors. CML stem cells/progenitors showed increased expression and activation of aPKCι (Fig. 1a, b and Supplementary Figure 1C) as compared to healthy donor CD34[+] cells. Leukemic B-cell progenitors derived from BCR-ABL[+] B-ALL patients showed enhanced aPKCι expression in comparison to healthy donor bone marrow (BM) derived B-cell progenitors (Supplementary Figure 1D). B-cell progenitors from both healthy donors and B-ALL patients express very reduced levels of aPKCζ compared with aPKCι (Supplementary Figure 1E). Interestingly, B-cell progenitors from B-ALL patients expressed ~2.5-fold higher levels of activated aPKCι, in comparison to normal BM B-cell progenitors (Fig. 1c, d). To examine whether BCR-ABL induces activation of aPKCλ in murine hematopoietic stem cells and progenitors (HSC/P), we transduced murine BM cells with p210-BCR-ABL retroviruses. BCR-ABL expression in murine BM Lin−c-kit+Sca1+ (LSK) cells led to significant upregulation and activation of aPKCλ (Fig. 1e, f). Like their human counterparts, murine stem cells and progenitors expressed high level of aPKCλ and very low levels of expression of aPKCζ (Supplementary Figure 1F). We identified a direct interaction between aPKCλ and p210-BCR-ABL by immunoprecipitation of aPKCλ in the cell lysates of transduced murine progenitors (Fig. 1g).

**aPKCλ is required for BCR-ABL induced leukemogenesis.** We hypothesize that the increased expression and activation of aPKCλ is essential for the maintenance and/or progression of p210-BCR-ABL[+] leukemia. To test this in vivo, we used a sequential transgenic animal model with inducible expression of p210-BCR-ABL upon doxycycline withdrawal (Dox-Off), and the deletion of aPKCλ upon poly I:C administration. To this end, we crossed Scl/p210 mice[26] with Mx1-Cre; aPKCζ−/−; aPKCλF/F mice[27] to generate the Scl/p210; Mx1-Cre; aPKCζ−/−; aPKCλF/F transgenic mouse model (Supplementary Figure 2A).

BM cells from Scl/p210; Mx1Cre; WT (WT), Scl/p210; Mx1Cre; aPKCζ−/− (aPKCζ−/−) Scl/p210; Mx1Cre; aPKCλF/F and Scl/p210; Mx1-Cre; aPKCζ−/−; aPKCλF/F were transplanted into primary BoyJ (B6.SJL.Ptprcᵃ.Pepcᵇ ᴮᵒʸᴶ) recipients to generate CD45.2[+] chimeric mice with nearly 100% donor chimera (Supplementary Figure 2B). Doxycycline administration was discontinued (Dox-off) to induce p210-BCR-ABL expression. Four weeks post BCR-ABL induction, chimeric mice had similar levels of myeloid, B cell and T cell chimera in PB (Supplementary Figure 2C). Poly I:C administration led to the deletion of the floxed alleles of aPKCλ[27]. After aPKCλ deletion (Supplementary Figure 2D), four groups of mice with WT or deletion (Δ/Δ) of aPKCλ were generated and analyzed, and named as Scl/p210;Mx1-Cre; wild type (WT), Scl/p210;Mx1-Cre; aPKCζ−/− (aPKCζ−/−); Scl/p210; Mx1-Cre; aPKCλΔ/Δ (aPKCλΔ/Δ), and Scl/p210; Mx1-Cre; aPKCζ−/−; aPKCλΔ/Δ (double knock-out, DKO).

Chimeric mice belonging to the WT and aPKCζ−/− groups developed CML-like disease characterized by leukocytosis, splenomegaly, hepatomegaly, and increased migration of HSC/Ps from the BM to the extramedullary tissues (Supplementary Figures 2E-I). Twenty-five percent of these mice died of the disease when followed up to six months post p210-BCR-ABL induction (Supplementary Figure 2E). The surviving 75% of the

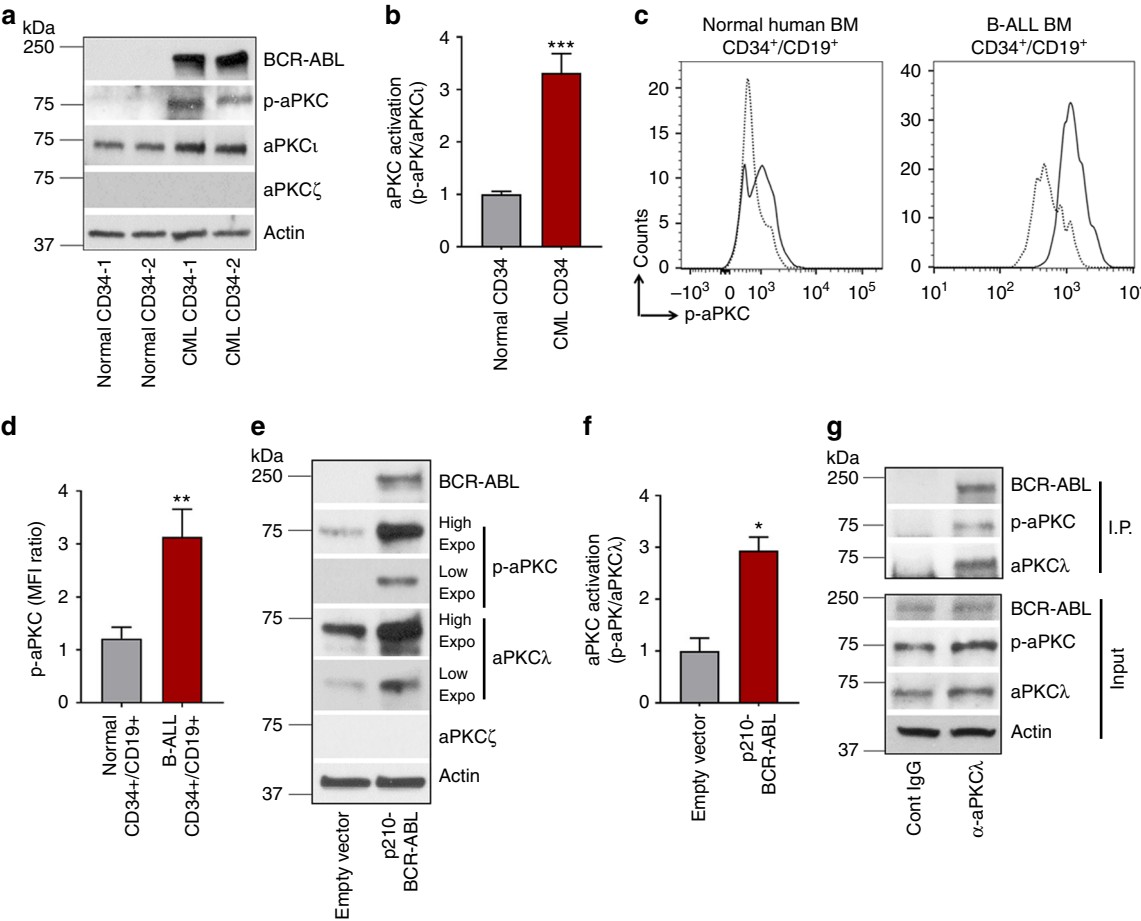

**Fig. 1** Human CML and B-ALL patients derived stem cells and progenitors (HSC/P) express aPKCι predominantly. **a** Representative example of western blot analysis of BCR-ABL, phospho-aPKC, aPKCι, aPKCζ, and actin in HSC/P (CD45+/CD34+) derived from healthy donors and CML patients mobilized peripheral blood (PB). 5 μg of whole cell lysate was loaded into each lane. **b** Western blot band intensity ratio between phospho-aPKCι and total aPKC expression. Atypical protein kinase Cι is predominantly expressed, and CML CD34+ cells show an increased level of activated aPKCι (phospho-aPKC) in comparison to healthy donor-derived CD34+ cells. **c, d** Histogram plot (**c**) and quantitation of phospho-aPKCι (**d**) as evidenced by phospho-flow analyses in B progenitors (CD45+/CD34+/CD19+) derived from healthy donor BM ($n = 2$) and B-ALL patients ($n = 3$). B-ALL patient's derived B-cell progenitors show enhanced p-aPKCι level in comparison to healthy donor-derived counterparts. **e** Representative example of western blot analysis of BCR-ABL, phospho-aPKCλ, aPKCλ, aPKCζ, and actin in mock (empty vector) and BCR-ABL-transduced murine HSC/P. **f** Western blot band intensity ratio between phospho-aPKCλ and total aPKC expression. **g** Representative western blots of BCR-ABL, p-aPKCλ, aPKCλ, and actin in the input and α-aPKCλ antibody immunoprecipitated (IP) fraction. Western blots with different exposure time for aPKCλ and phospho-aPKCλ were presented for quantification accuracy; Low (15 sec), high (1 min). Cont: control. Data are presented as mean ± SD of a minimum of 2 independent experiments. *$p < 0.05$; **$p < 0.01$, $t$-test

mice in the WT and aPKCζ−/− groups also showed a CML-like disease phenotype when analyzed at autopsy. However, none of the aPKCλΔ/Δ or DKO chimeric mice died (Supplementary Figure 2E). The spleen weight, PB neutrophil and both PB and spleen progenitor counts (LSK and colony forming units, CFU) of aPKCλΔ/Δ and DKO chimeric mice were reduced (Supplementary Figure 2F-K). Interestingly, we observed no changes in the frequencies of BM LSK cells in aPKC deficient chimeric mice (Supplementary Figure 2L).

The expression and activation of p210-BCR-ABL in WT or aPKC deficient HSC/Ps were similar (Supplementary Figure 2M). Of note, the expression of aPKCζ and aPKCλ at the protein levels in the stem cells and progenitors derived from this animal model (Supplementary Figures 3A–B) gave a pattern similar to that observed in primary leukemic cells (Fig. 1 and Supplementary Figure 1). Taken together, these data suggest that aPKCλ collaborates in the development of the splenic and blood hallmarks of p210-BCR-ABL induced CML-like disease.

**aPKCλ deficiency abrogates BCR-ABL+ B-ALL.** BCR-ABL+ leukemia progresses from a chronic phase to a poor prognosis blast crisis phase, in which myeloid or lymphoid blasts fail to differentiate[2–4]. To determine the role of aPKCs in the blast crisis phase of BCR-ABL+ leukemia, BM cells from primary recipient leukemic mice along with an equivalent number of CD45.1+ competitor cells (BoyJ BM cells) were transplanted serially into secondary and tertiary BoyJ recipients. Mice were kept on a *Dox-Off* diet to induce BCR-ABL expression. WT and aPKCζ−/− secondary chimeric mice developed B-ALL with median survival of 61.5 days and 52.5 days, respectively (Fig. 2a). aPKCζ−/− chimeric mice died significantly earlier than the WT group ($p < 0.05$, $t$-test). Strikingly, aPKCλΔ/Δ and DKO chimeric mice survived up to 150 days without any sign of disease ($p < 0.001$, $t$-test; Fig. 2a). At 6 and 8 weeks, the PB CD45.2+ leukemic chimera were significantly decreased in the aPKCλΔ/Δ and DKO groups (Fig. 2b). These results were not due to a reduced inoculum of BM progenitors in the graft, since the frequencies of LSC/P in the BM from the aPKCλΔ/Δ or DKO chimeric mice were similar to

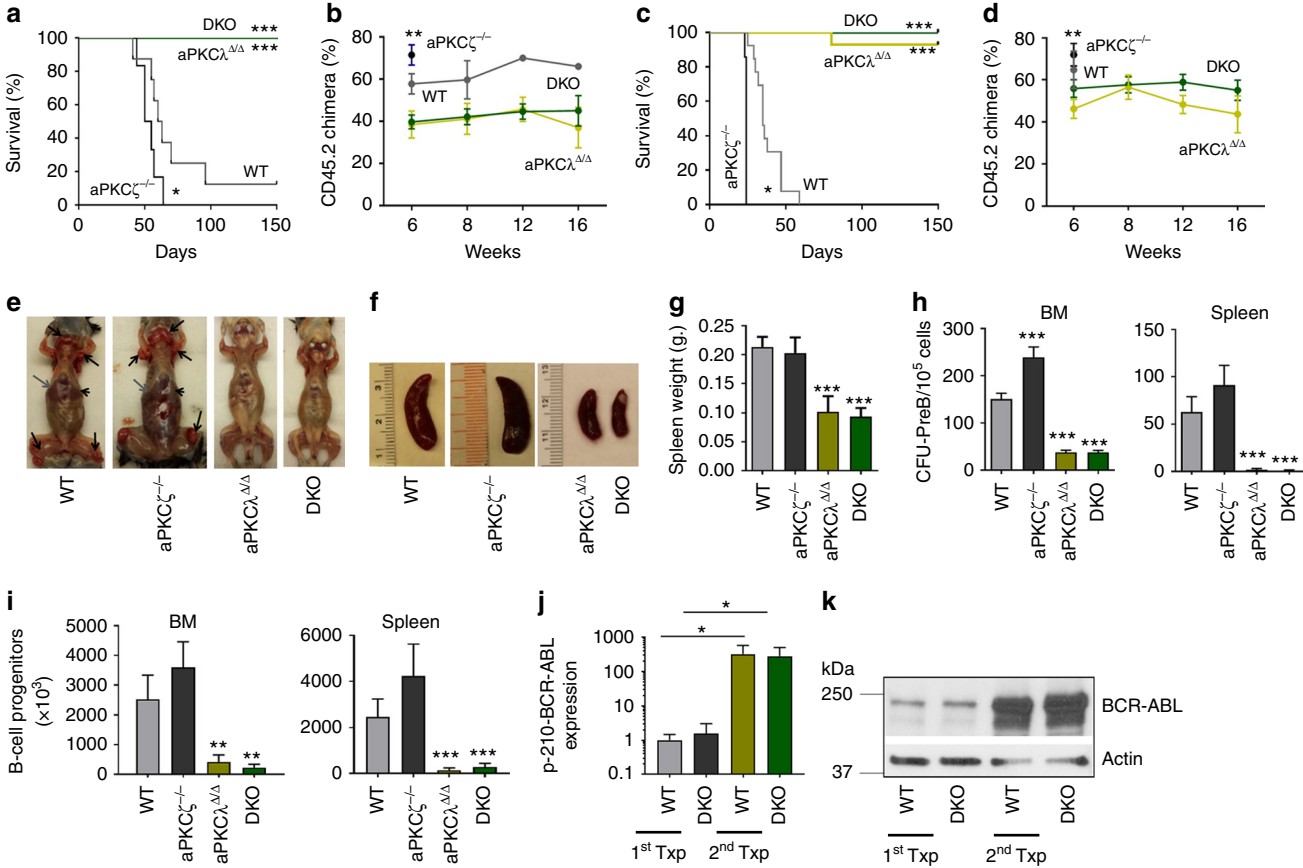

**Fig. 2** Deficiency of aPKCλ, but not aPKCζ, abrogates the development of B-ALL. **a** Kaplan–Meier survival analyses of competitively transplanted secondary recipients. Whole BM cells derived from Scl/p210: WT, aPKCζ$^{-/-}$, aPKCλ$^{Δ/Δ}$ and DKO chimeric primary recipients along with competitor cells (CD45.1$^{+}$ BoyJ BM cells) were transplanted into B6.SJL.Ptprc$^a$.Pepc$^{b\ BoyJ}$ (BoyJ) secondary recipients at 1:1 ratio. Mice were followed for the development of leukemia, and sick and moribund mice were sacrificed. **b** Donor leukemic chimera (CD45.2$^{+}$ cells) in the PB of secondary recipients at different time interval after transplantation. **c** Kaplan–Meier survival analyses of competitively transplanted tertiary recipients. Whole BM cells from Scl/p210; WT, aPKCζ$^{-/-}$, aPKCλ$^{Δ/Δ}$ and DKO chimeric secondary recipients transplanted into BoyJ tertiary recipients, and the development of leukemia was followed. **d** Donor leukemic chimera (CD45.2$^{+}$ cells) in the PB of tertiary recipients at different time interval after transplantation. **e** Lymphadenopathies (black arrows), hepatomegaly (gray arrows) and splenomegaly (arrowheads) in WT and aPKCζ$^{-/-}$ chimeric secondary recipient mice. aPKCλ$^{Δ/Δ}$ and DKO chimeric mice do not show any signs of B-ALL when followed up to 6 months post-transplantation **f, g** Representative example of spleen images (**f**) and spleen size (**g**) of Scl/p210; WT, aPKCζ$^{-/-}$, aPKCλ$^{Δ/Δ}$ and DKO chimeric mice at autopsy. **h** CFU-pre B content in the BM and spleen. **i** FACS quantification of leukemic B-cell progenitors in the BM and spleen of secondary recipient chimeric mice at autopsy. Deficiency of aPKCλ results in drastic reduction in leukemic B-cell progenitors content in the BM and spleen. **j** Q-RT-PCR analyses of p210-BCR-ABL (B3A2) expression in B progenitors derived from primary and secondary recipient chimeric mice. B-cell progenitors derived from secondary recipients show significant upregulation of p210-BCR-ABL expression. **k** Representative example of western blot of p210-BCR-ABL expression in B-cell progenitors derived from primary and secondary recipient chimeric mice. Data are presented as mean ± SD of a minimum of three independent experiments. $^*p < 0.05$; $^{**}p < 0.01$; $^{***}p < 0.001$, t-test

their WT and aPKCζ$^{-/-}$ respective counterparts (Supplementary Figure 2L). These survival curves in secondary recipients were confirmed in tertiary recipients, with WT and aPKCζ$^{-/-}$ chimeric mice showing aggressive leukemia and increased mortality with a median survival of 35 and 24 days, respectively (Fig. 2c). However, aPKCλ$^{Δ/Δ}$ and DKO chimeric tertiary recipient mice survived disease-free when followed up to 5 months post-transplantation (Fig. 2c). PB CD45.2$^{+}$ leukemic chimera were significantly decreased in the aPKCλ$^{Δ/Δ}$ and DKO groups (Fig. 2d). The PB chimerisms in the aPKCλ$^{Δ/Δ}$ and DKO groups were maintained at similar levels (Fig. 2b, d). WT and aPKCζ$^{-/-}$ chimeric mice displayed splenomegaly, lymphadenopathy, head and neck tumor formation, and hind-limb paralysis secondary to central nervous system infiltration of leukemic cells (Fig. 2e–g). The BM and spleen CFU-Pre-B and immunophenotypically identified B-cell progenitors (CD45.2$^{+}$FSC$^{hi}$B220$^{lo}$CD19$^{+}$ D43$^{+}$IgM$^{-}$) contents in aPKCλ$^{Δ/Δ}$ and DKO chimeric mice were

significantly decreased, indicating that the deficiency of aPKCλ, or both aPKCλ and –ζ, impairs the maintenance of leukemic B-cell progenitors (Fig. 2h, i).

While the expression of p210-BCR-ABL mRNA in BM myeloid progenitors (Lin$^{-}$ cKit$^{+}$, LK cells) derived from primary and secondary recipient mice was similar (Supplementary Figure 3C), we observed a dramatic upregulation of p210-BCR-ABL, both at the mRNA and protein levels, in B-cell progenitors derived from secondary recipient B-ALL mice (Fig. 2j, k). These results strongly suggest that the selection of high BCR-ABL-expressing cells is likely to be responsible for the aggressive B-ALL seen in the secondary and tertiary recipient mice. Although the upregulation of BCR-ABL expression is similar in both WT and DKO B-cell progenitors (Fig. 2j, k), aPKCλ$^{Δ/Δ}$ and DKO chimeric mice did not develop aggressive B- ALL, suggesting that aPKCλ is required for the development of BCR-ABL induced B-ALL.

**aPKCλ deletion impairs leukemic B-cell progenitor activity**. To understand the molecular mechanisms whereby aPKCλ contributes to B-ALL development, we performed comparative whole transcriptome RNA-seq analyses of total RNA isolated from WT and DKO leukemic B-cell progenitors ($n = 3$ mice for each group). The expression of a total of 2448 genes (2172 upregulated and 276 downregulated) in DKO leukemic B-cell progenitors was differentially regulated (<0.8 to >1.2-fold, $p < 0.05$; Supplementary Data 1 [https://www.ncbi.nlm.nih.gov/geo/query/acc.cgi?acc=GSE121018]) in comparison to WT counterparts. Gene ontology (GO) and pathway analysis identified differential expression of genes involved in proliferation, cell cycle progression, B-cell differentiation, histone modification and chromatin organization as major contributors to the differences between WT and DKO leukemic B-cell progenitors (Fig. 3a, Supplementary Data 2 [https://www.ncbi.nlm.nih.gov/geo/query/acc.cgi?acc=GSE121018]). Interestingly, the expression of *Prkcb1* and *Prkcq* was upregulated in DKO group, with no significant changes in other PKC isoforms (Supplemental Table 1). The upregulation of *Prkcb1* mRNA expression did not translate into increased protein levels. However, PKCθ level is increased in aPKCλ$^{Δ/Δ}$ and DKO cells and decreased in aPKCζ$^{−/−}$ progenitors, which is consistent with a possible tumor suppressor role of PKCθ (Supplementary Figure 3D).

The progression of BCR-ABL$^+$ leukemia is associated with genetic instability, increased proliferation, reduced apoptosis, and differentiation arrest[2–5]. Therefore, we examined proliferation, apoptosis, and differentiation of aPKC-proficient or -deficient BCR-ABL$^+$ leukemic B-cell progenitors. The result of an in vivo 5-bromo-2′-deoxyuridine (BrdU) incorporation assay showed a reduced proliferative potential of aPKCλ$^{Δ/Δ}$ and DKO cells (Fig. 3b). This is accompanied by reduced *Ccnd1* (cyclin-D1), and increased *Cdkn1a* (p21), *Cdkn1b* (p27) levels in aPKCλ$^{Δ/Δ}$ or DKO cells (Supplementary Figure 3E). Furthermore, aPKCλ$^{Δ/Δ}$ and DKO leukemic B-cell progenitors showed increased apoptosis (Fig. 3c). In contrast, deletion of aPKCζ alone significantly increased the proliferation and survival of leukemic B-cell progenitors (Fig. 3b, c).

To determine the molecular mechanisms underlying these functional alterations, we analyzed the effect of aPKC deficiency on previously reported bona fide signaling pathways downstream of BCR-ABL[28]. The activation of p42/44 MAPK (ERK1/2), and its upstream kinase, MEK1/2, were dramatically impaired in leukemic B-cell progenitors lacking either aPKCλ or both aPKCζ and -λ isoforms without major changes in aPKCζ$^{−/−}$ cells (Fig. 3d). No changes were found in the levels or activities of p38, Jnk, or NF-κB among the groups (Supplementary Figure 3F). We found no change in the activation of c-Raf or B-Raf, upstream kinases of MEK1/2 (Supplementary Figure 3G) suggesting a Raf-independent mechanism of MEK/ERK activation. Furthermore, deficiencies of either aPKCλ or both aPKC isoforms resulted in impaired activation of Rac GTPase with no significant changes in the levels of activated Ras GTPase (Fig. 3e and Supplementary Figure 3H). The decreased Rac GTPase activation could explain the downregulation of the MEK1/2-ERK1/2 MAPK pathway, as Rac GTPase is a major regulator of MEK/ERK activation[29]. To rule out any compensatory effect in vivo, we transduced hematopoietic progenitors from aPKCλ$^{F/F}$ mice with an empty vector or Cre-recombinase expressing retroviruses to induce the deletion of aPKCλ in vitro. Consistent with our in vivo data, the in vitro deletion of aPKCλ led to reduced MEK/ERK activation (Supplementary Figure 3I).

To further understand how aPKCλ is activated, we examined the level of Cdc42 activation, the upstream regulator of aPKCs. The aging of HSC and transformation of HSC/P to AML are associated with increased CDC42 expression and activity[30,31]. Indeed, BCR-ABL expression led to the upregulation of Cdc42

activity (Supplementary Figure 4A). The pharmacological inhibition of Cdc42 by casin abrogated the activation of aPKCλ (Supplementary Figure 4B). Further, leukemic progenitors treated with imatinib showed decreased aPKCλ activation (Supplementary Figure 4C). These results, taken together, demonstrate that Cdc42 controls the activation of aPKCλ in BCR-ABL$^+$ leukemic progenitors. The biochemical data obtained from our mouse genetic model indicates that aPKCλ regulates the Rac GTPase mediated activation of MEK1/2-ERK1/2 MAPK pathway (Fig. 3d, e). To validate this, we treated primary leukemic progenitors with NSC23766 (a specific Rac GTPase inhibitor) and analyzed MEK1/2-ERK1/2 activation. Pharmacological inhibition of Rac GTPase impaired the activation of MEK1/2-ERK1/2 MAPK pathway (Supplementary Figure 4D).To further identify the prevalent mechanism downstream of aPKCλ dependent MEK1/2-ERK1/2 activation, we treated BCR-ABL$^+$ leukemic B-cell progenitors with the MEK inhibitor trametinib or the Raf inhibitor debrafenib. Trametinib treatment completely abrogated the MEK1/2-ERK1/2 MAPK pathway activation (Supplementary Figure 4E). However, debrafenib significantly attenuated but did not abrogate MEK1/2-ERK1/2 activation (Supplementary Figure 4F), suggesting that aPKCλ-Rac GTPase mediated MEK1/2-ERK1/2 activation is the predominant pathway in these leukemic progenitors.

Gene set enrichment analysis (GSEA) using the Molecular Signatures Database (MSigDB v4.0) revealed that among the most significantly enriched (false discovery rate <0.001) were gene sets associated with the suppression of the transcriptional activity of Myc in DKO cells (Supplementary Figure 5A). We found that Myc was highly upregulated in aPKCζ$^{−/−}$ cells and downregulated with the deletion of aPKCλ (Supplementary Figures 5B–D). However, Myc expression in DKO leukemic B-cell progenitors was intermediate between WT and aPKCλ$^{Δ/Δ}$ cells, indicating that aPKCζ and -λ have antagonistic roles in regulating Myc expression. The expression of Myc was regulated post-translationally, as we found no significant changes at mRNA levels (Supplementary Figure 5E). Several mechanisms controlling Myc activation and stabilization have been reported including MEK/ERK-dependent phosphorylation (reviewed in[32]). To examine if aPKCλ dependent MEK/ERK activation regulates Myc protein stability, we treated WT, aPKCζ$^{−/−}$ and DKO leukemic B-cell progenitors with the MEK inhibitor trametinib. The inhibition of MEK/ERK MAPK pathway led to rapid (~80%) and late (~20%) decreases in Myc protein expression in WT and aPKCζ$^{−/−}$ cells, respectively (Supplementary Figure 5F-G). However, DKO leukemic cells which are defective in MEK/ERK activation were, as expected, unresponsive to MEK inhibition. Altogether, in this cellular context, aPKCζ acts as a tumor suppressor by negatively regulating Myc expression, as shown in other cancers[33].

**Deficiency of aPKCλ induces leukemic B-cell differentiation**. B-cell differentiation arrest is a pivotal event in the development of B-ALL[2–5]. The immunophenotypic characterization of BM cells in serially transplanted chimeric mice showed an accumulation of B-cell progenitors (CD45.2$^+$B220$^{lo}$CD19$^+$CD43$^+$IgM$^−$) accompanied by a dramatically diminished number of preB cells (CD45.2$^+$B220$^{lo}$CD19$^+$CD43$^{−/lo}$IgM$^−$) and mature/immature B cells (CD45.2$^+$B220$^{hi}$CD19$^+$CD43$^−$IgM$^+$) in WT and aPKCζ$^{−/−}$ chimeric mice (Fig. 3f and Supplementary Figure 6A). Similar trends were seen in the splenic and PB contents of leukemic B-cell progenitors (Supplementary Figures 6A–C). However, a deficiency of aPKCλ or both aPKCζ and -λ reversed the leukemic B cell differentiation arrest (Fig. 3f, Supplementary Figures 6A–C), indicating that

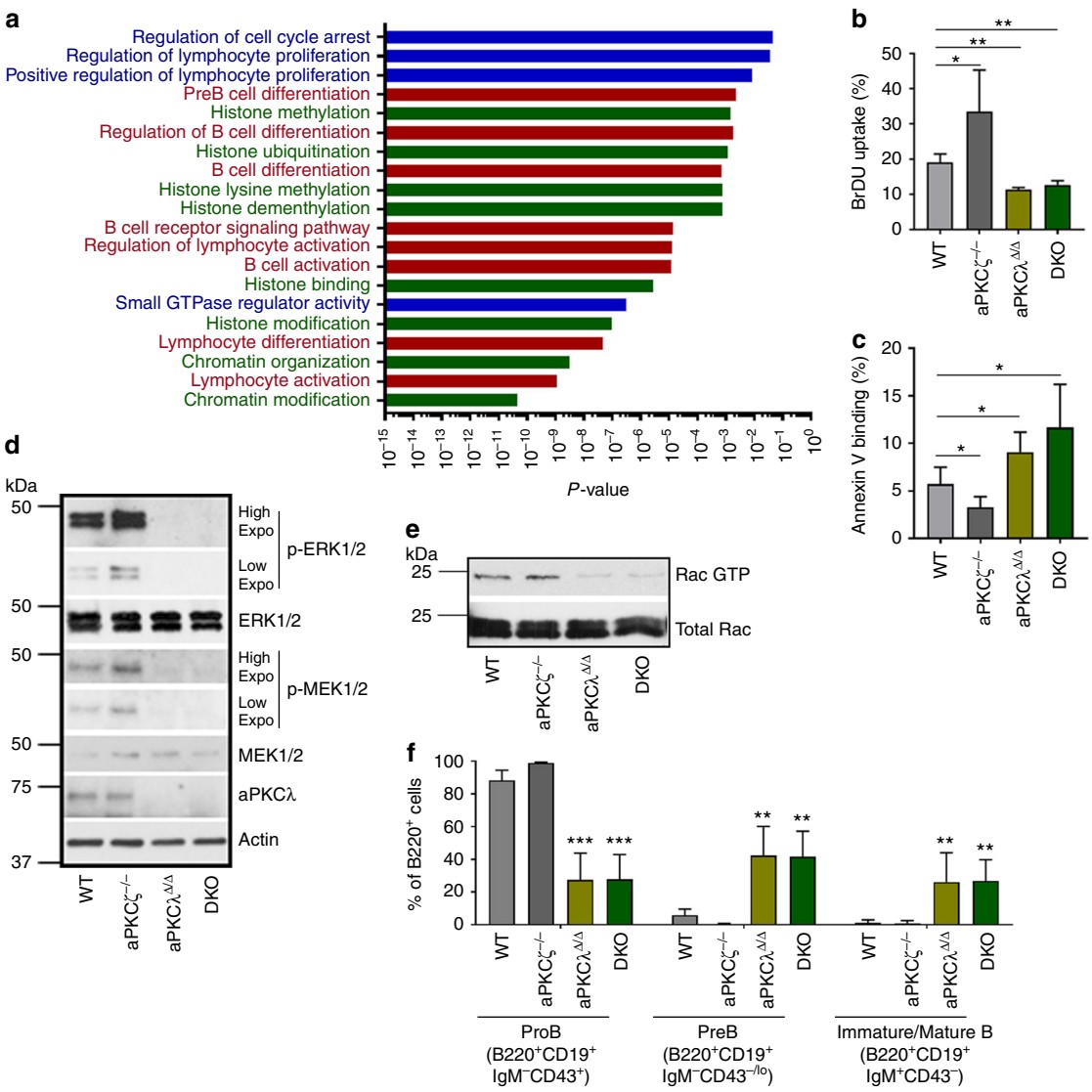

**Fig. 3** aPKCλ deficiency impairs proliferation, survival and B cell differentiation arrest. **a** Comparative transcriptome and gene-ontology (GO) pathway analyses of the differential expression of genes in WT and DKO leukemic B-cell progenitors showing the differential regulation of genes involved in proliferation, cell cycle regulation, B cell differentiation network, and histone and chromatin modifications. Pathways shown in Blue - proliferation and cell cycle regulation; in Red - B cell differentiation network; in Green - histone and chromatin modification. **b** In vivo BrDU uptake by leukemic B-cell progenitors in Scl/P210; WT, aPKCζ$^{-/-}$, aPKCλ$^{Δ/Δ}$, and DKO chimeric mice. **c** FACS quantification of annexin V-binding of WT, aPKCζ$^{-/-}$, aPKCλ$^{Δ/Δ}$ and DKO leukemic B-cell progenitors. **d** Representative example of western blots of p-ERK1/2, ERK1/2, p-MEK1/2, MEK1/2, aPKCλ, and Actin in WT, aPKCζ$^{-/-}$, aPKCλ$^{Δ/Δ}$, and DKO leukemic B-cell progenitors. Activation of MEK/ERK MAPK pathway is impaired in aPKCλ deficient leukemic B-cell progenitors. Western blots with different exposure time for phospho-Erk1/2 and phospho-Mek1/2 were presented to show minimal expression; Low (15 sec), high (1 min). **e** Representative example of the analyses of Rac GTPase activation by specific effector pulldown (PAK-PBD agarose) assay in leukemic B-cell progenitors derived from WT, aPKCζ$^{-/-}$, aPKCλ$^{Δ/Δ}$ and DKO chimeric mice. **f** FACS-quantification of proB, preB and immature/mature B cells in the BM of WT, aPKCζ$^{-/-}$, aPKCλ$^{Δ/Δ}$ and DKO chimeric mice. Data are presented as mean ± SD of a minimum of three independent experiments. *$p < 0.05$; **$p < 0.01$; ***$p < 0.001$, $t$-test

aPKCλ is required for B cell differentiation arrest in BCR-ABL$^+$ leukemia.

Gene ontology (GO) and pathway analysis of the RNAseq data identified the upregulation of genes involved in the B-cell differentiation network, such as *Pax5, Ikzf3, Rag1* and *Rag2* and the downregulation of the prosurvival gene *Bcl3* (Fig. 4a). The enhanced expression of B-cell differentiation genes (*Pax5, Ebf1, Ikzf1, Ikzf3, Rag1*) were further validated by Q-RT-PCR (Fig. 4b, c and Supplementary Figures 6D–F). Deficiency of aPKCλ or both aPKCζ and -λ led to the increased expression of the cell cycle inhibitor *Cdkn2a* (Fig. 4d). Interestingly, the expression of stem cell self-renewal genes (*Scl/Tal1, Runx1, Hmgb3, Cbx5*) remained unchanged (Supplementary Figures 6G–J).

We examined the effect of aPKCζ and-λ deficiencies in normal B-cell differentiation in an inducible murine model (Mx1-Cre; aPKCζ$^{-/-}$; aPKCλ$^{F/F}$). In accordance with previously published data from our laboratory[27], hematopoietic lineage distribution (myeloid, B-cell, and T-cell) remained unchanged in the absence of aPKCλ and/or aPKCζ (Supplementary Figure 7A). A deficiency of aPKCλ and/or aPKCζ did not affect normal B-cell lineage maturation (Supplementary Figures 7B–C). The survival of normal B-cell progenitors, pre-B, and immature/mature B cells was not affected by the deletion of aPKCs (Supplementary Figure 7D). Also, a deficiency of aPKCλ or both isoforms did not affect the frequency of BM colony forming unit (CFU)-pre-B content (Supplementary Figure 7E). The expression levels of

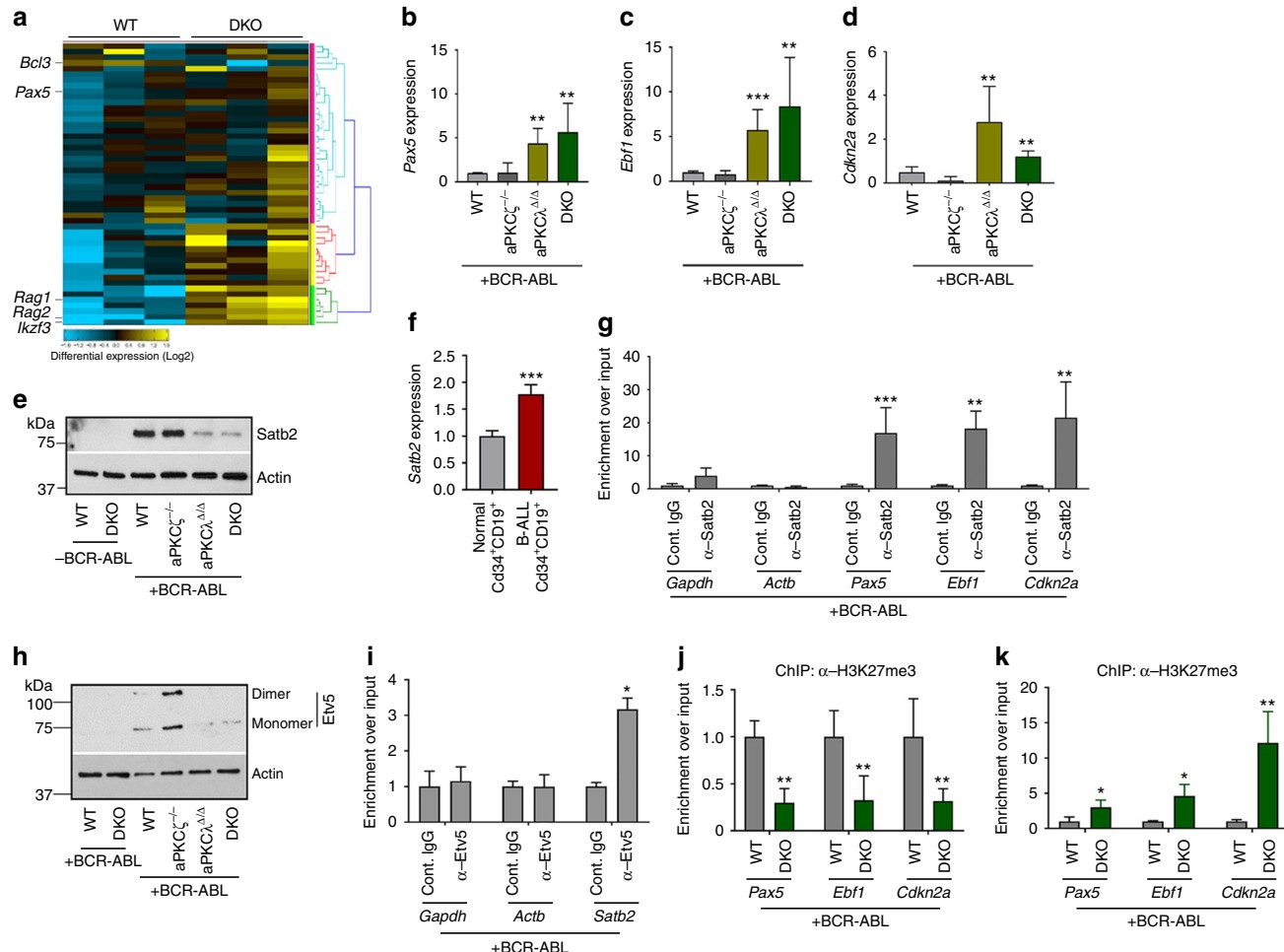

**Fig. 4** Satb2 and Etv5 downstream of aPKCλ regulates B cell differentiation arrest. **a** Heat map of the B-cell differentiation network in WT and DKO leukemic B-cell progenitors whole transcriptome. **b–d** Q-RT-PCR analyses of genes [*Pax5* (**b**), *Ebf1* (**c**), *and Cdkn2a* (**d**)] that are involved in B-cell differentiation program in leukemic B-cell progenitors derived from WT, aPKCζ$^{-/-}$, aPKCλ$^{Δ/Δ}$, and DKO chimeric mice. **e** Representative example of western blots showing the expression of chromatin modifier, Satb2, and actin in WT, aPKCζ$^{-/-}$, aPKCλ$^{Δ/Δ}$ and DKO leukemic B-cell progenitors. **f** Q-RT-PCR analysis of *SATB2* in human BM B progenitors (CD34$^+$/CD19$^+$) from normal individuals ($n = 2$) and Ph$^+$ B-ALL pediatric patients ($n = 3$). **g** Chromatin immunoprecipitation using α-Satb2 antibody and Q-PCR analyses of genes involved B-cell differentiation program *Pax5, Ebf1,* and *Cdkn2a* in p210-BCR-ABL expressing leukemic B-cell progenitors. Leukemic B-cell progenitors show enhanced enrichment of *Pax5, Ebf1,* and *Cdkn2a* loci. We found no enrichment of the housekeeping genes *Gapdh* and *Actb*. **h** Representative example of western blots showing expression of Etv5 in WT, aPKCζ$^{-/-}$, aPKCλ$^{Δ/Δ}$, and DKO leukemic B-cell progenitors. **i** Chromatin immunoprecipitation using Etv5 antibody and Q-PCR analyses of Stab2 promoter enrichment in leukemic B-cell progenitors. We found no enrichment of the housekeeping genes *Gapdh* and *Actb*. **j–k** Chromatin immunoprecipitation using α-H3K27me3 (**j**) and α-H3K4me3 (**k**) antibodies and Q-PCR analyses of the enrichment of *Pax5, Ebf1* and *Cdkn2a* genes in WT and aPKC deficient B-cell progenitors. DKO leukemic B-cell progenitors show enhanced activation of genes involved in B-cell differentiation in comparison to their WT counterparts. Cont: Control. Data are presented as mean ± SD of a minimum of two independent experiments. *$p < 0.05$; **$p < 0.01$; ***$p < 0.001$, t-test

*Pax5, Ebf1, Ikzf1, Ikzf3,* and *Rag1* remained unaffected in aPKCλ$^{Δ/Δ}$ and DKO B-cell progenitors (Supplementary Figure 7F). Altogether, these data indicate that the survival, proliferation, and differentiation fate of lymphoid leukemic, but not normal, B-cell progenitors depends on aPKCλ activity.

**aPKCλ deficiency induces Numb nuclear translocation.** Activated aPKC phosphorylates cell fate determinants Numb and Lgl1, regulates their asymmetric distribution, and controls cell differentiation[17]. Numb also translocates to the nucleus and acts as a p53 stabilizer. Deletion of aPKCλ, or both isoforms, led to reduced phosphorylation, impaired cytoplasmic polarization, and nuclear translocation of Numb, without modifying the expression at the protein or mRNA levels (Supplementary Figures 8A–E). In addition, the loss of aPKCλ led to a significant reduction in the

phosphorylation of Lgl1, a component of the basolateral polarity complex (Supplementary Figures 8F–G). Together, these data suggest that the loss of cytoplasmic Numb/Lgl1 polarization and activation may mediate increased B-cell differentiation in the absence of aPKCλ or both isoforms. However, p53 expression at protein levels in leukemic (or non-leukemic) cells was not affected by the loss of aPKCs (Supplementary Figures 8H), indicating that at least in these cells, p53 expression was insensitive to nuclear Numb expression.

**Satb2, downstream of aPKCλ, controls B cell differentiation.** Differential expression analysis of known epigenetic regulators highlighted the differential expression of the chromatin modifiers Satb1 (unchanged) and Satb2 (downregulated) in DKO leukemic B-cell progenitors when compared with their WT counterpart, as

validated by Q-RT-PCR (Supplementary Figure 9A–B) and western blot analyses (Fig. 4e). Interestingly, deletion of aPKCs did not modify the *Satb2* or *Satb1* expression in non-leukemic B-cell progenitors at the mRNA (Supplementary Figure 9A–B) and protein levels (Fig. 4e), which were very low for *Satb2*. The effect of aPKCλ deficiency on Satb2 expression depended on the induction of BCR-ABL, which upregulated Satb2 by as much as ~10-fold, indicating that oncogenic signaling mediated by aPKCλ is required for Satb2 transcriptional upregulation. The upregulation of *SATB2* in BCR-ABL[+] leukemic B-cell progenitors was confirmed in human B-ALL CD34[+]/CD19[+] cells (Fig. 4f). Together, these data identified Satb2 as a possible mediator of aPKC-dependent leukemogenesis in BCR-ABL[+] B-cell progenitors.

The special adenine-thymine (AT)-rich sequence-binding proteins Satb1 and Satb2 have been reported as chromatin modifiers that regulate gene expression and higher order chromatin structure of multigene clusters[34,35]. Furthermore, Satb2 has been described as a transcriptional repressor[35,36]. Our data indicate that the expression of Satb2 and a network of B-cell proliferation and differentiation genes (including *Pax5*, *Ebf1*, and Cdkn2a) associated with pre-B ALL are inversely correlated in leukemic B-cell progenitors. Based on these associations, we hypothesized that Satb2 expression and activity were mechanistically related with this transcriptionally regulated network. We analyzed the binding of Satb2 to the promoter regions of these genes using chromatin immunoprecipitation and Q-RT-PCR assays (ChIP-Q-PCR). B-cell progenitors derived from WT leukemic mice were subjected to ChIP using an anti-Satb2 antibody. Quantitative genomic analysis of Satb2 binding to the promoter regions of *Pax5*, *Ebf1*, and *Cdkn2a* demonstrated a significantly increased enrichment of these genes in leukemic B-cell progenitors (Fig. 4g), indicating that Satb2 was bound to and could possibly modulate the expression of these genes.

To investigate how the expression of *Satb2* is regulated at the transcriptional level in leukemic B-cell progenitors, we looked for the presence of a consensus motif for specific transcription factors in the promotor region of the *Satb2* gene. Interestingly, the *Satb2* promoter region contains several bona fide core consensus motifs for the Ets family transcription factor [5′-GGA(A/T)-3′][37]. Transcriptome analyses of WT and aPKCs deficient leukemic B-cell progenitors trended towards a downregulation of the expression of PEA3 (polyomavirus enhancer activator 3) subfamily members of the Ets family of transcription factors, such as *Etv5* and *Etv1* along with *Satb2*, but not of other members of the Ets family (Supplementary Figure 9C). PEA3 transcription factors Etv1, Etv4, and Etv5 overexpression is associated with the development and metastasis of prostate cancer, colorectal cancer, and ovarian cancer[38,39]. The activation of the ERK MAPK pathway positively regulates the expression of Etv1, Etv4, and Etv5[40]. Etv5, in particular, has been identified as a major regulator of ERK-dependent transcriptional regulatory networks in BCR-ABL[+] leukemogenesis[41,42]. Since ERK activation is dramatically impaired in aPKC deficient leukemic B-cell progenitors (Fig. 3d), we examined Etv5 expression in the presence and absence of aPKCλ in the context of p210-BCR-ABL expression. Etv5 expression was significantly upregulated, along with Etv5 dimerization in leukemic cells, and the deficiency of either aPKCλ or both aPKC isoforms reversed it (Fig. 4h and Supplementary Figure 9D). The Etv5 homodimer possesses higher affinity for DNA binding than the Etv5 monomer[43], and leukemic B-cell progenitors lacks the active homodimer Etv5 in the absence of aPKCλ. However, Etv5 expression remained unchanged in aPKCs deficient B-cell progenitors in the absence of BCR-ABL expression (Fig. 4h and Supplementary Figure 9D). To further validate that MEK1/2-ERK1/2 MAPK pathway controls

Etv5 expression, we treated leukemic cells with the MEK inhibitor trametinib and Raf inhibitor debrafenib. Pharmacological inhibition of MEK reduced the expression of *Etv5* by more than 90%, whereas, Raf inhibition did not significantly affect it (Supplementary Figures 9E–F). To further demonstrate that Etv5 transcriptionally regulates Satb2 expression, WT leukemic cells were subjected to α-Etv5 ChIP assay. We found significant enrichment of *Satb2* promoter regions in the α-Etv5 Ab chromatin immunoprecipitate fraction (Fig. 4i). Together, these data indicate that Etv5 binds to the Satb2 promoter and that the expression of Etv5 and Satb2 follow the same pattern upon expression of BCR-ABL and the aPKC isoforms.

To determine the epigenetic regulation of the expression of *Pax5*, *Ebf1*, and *Cdkn2a* in leukemic B-cell progenitors in the context of aPKC expression, we performed quantitative analyses of the enrichment of histone marks (activation and repression) of the promoter region of these loci by ChIP and Q-PCR analyses. There was a significantly reduced (~70% less) enrichment of H3K27me3 repression marks and an increased (3–10 fold) enrichment of H3K4me3 activation marks in the *Pax5*, *Ebf1*, and *Cdkn2a* loci in aPKC deficient leukemic B-cell progenitors in comparison to the WT counterpart (Fig. 4j, k). The downregulation of repression and upregulation of activating histone marks was in consistent with the previously reported role of Satb2 as a recruiter of histone deacetylase (HDAC) repression activity in neuronal function[44]. Together, these data demonstrate that the ERK, Etv5 and Satb2 signaling cascade downstream of aPKC regulates the differentiation arrest and transformation of BCR-ABL[+] B-cell progenitors.

**Etv5 or Satb2 expression rescue the differentiation arrest.** To determine whether our previous observations are mechanistically linked, and to examine if the reversal of leukemic B-cell differentiation induced by aPKCλ deficiency depends on the downregulation of Etv5 and Satb2, we performed a series of Etv5/Satb2 rescue experiments. We transduced LSK cells derived from WT or DKO primary leukemic mice with either a mock or Etv5 or Satb2-expressing lentiviral vectors, and analyzed B cell output after co-culturing the transduced cells on the OP9 stromal monolayer in vitro or in vivo after transplantation into secondary recipients (Fig. 5a). The forced expression of Etv5 rescued the expression of *Satb2* in aPKC deficient cells (Fig. 5b, c), and resulted in significantly increased B-cell progenitor generation (Fig. 5d and Supplementary Figure 10A). Furthermore, the lentiviral expression of Satb2 led to B-cell differentiation arrest and leukemogenesis as evidenced by the increased frequency of B-cell progenitors in the BM and spleen, significantly increased spleen size (Fig. 5e, f and Supplementary Figure 10B), enhanced CFU-preB content of BM (Fig. 5g), and restoration of the B-cell differentiation arrest in the BM and spleen (Fig. 5h, i). Transduction of LSK cells with a Satb2-expressing vector resulted in the reversal of leukemic B-cell progenitor Satb2 levels, but not of Etv5 (Fig. 5j, k), further validating that Satb2 expression is downstream of Etv5. Q-RT-PCR analyses of total RNA isolated from B-progenitors derived from mock/Satb2 transduced and transplanted secondary recipients further confirmed the effect of forced Satb2 expression on the reversal of the hallmark leukemogenic B-cell transcriptional signature (Fig. 5l–n and Supplementary Figure 10C). The lentiviral expression of Satb2 led to enhanced enrichment of H3K27me3 marks (Fig. 5o) and reduced levels of H3K4me3 marks (Fig. 5p) at *Pax5*, *Ebf1*, and *Cdkn2a* loci. These results demonstrate that Satb2, downstream of aPKCλ, functions as a chromatin repressor to inhibit the expression of genes required for B-cell maturation. Interestingly, forced expression of Satb2 did not rescue the increased apoptosis of

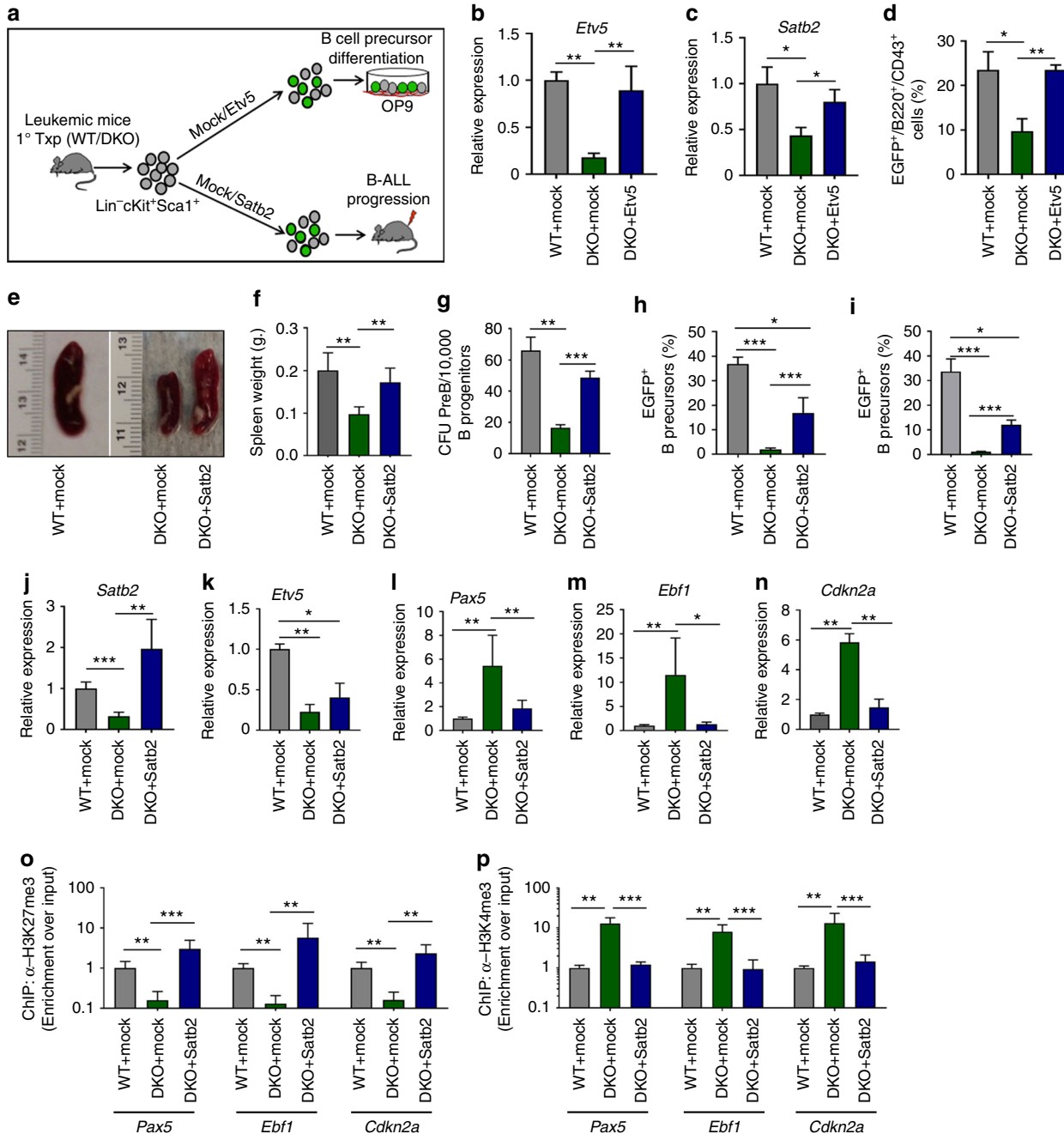

**Fig. 5** Forced expression of Etv5 and Satb2 rescue B cell differentiation arrest. **a** Schematic diagram of Etv5 and Satb2 ectopic expression in Lin−cKit+Sca1+ (LSK) cells derived from Scl/p210 primary chimeric mice and co-culture on a OP9 mouse stromal cells monolayer and/or in vivo transplantation into secondary recipients. **b** Q-RT PCR showing expression of *Etv5* in mock/Etv5 transduced B progenitors. **c** Q-RT PCR analyses showing rescue of Satb2 expression in Etv5 transduced B progenitors. **d** FACS quantification of EGFP+ (mock/Etv5/Satb2 expressing) B-cell progenitors derived from OP9 stromal monolayer co-culture of mock/Etv5 transduced LSK cells. **e, f** Representative spleen images (**e**) and spleen weight (**f**) of mock/Satb2 transduced and transplanted secondary recipient mice. **g** Satb2 ectopic expression enhances the CFU- preB content. **h, i** FACS quantification of EGFP+ B-cell progenitors content in the bone marrow (BM) (**h**) and spleen (**i**) of secondary recipient mice transplanted with mock/Satb2 transduced LSK cells. **j, k** Q-RT PCR analyses of Satb2 (**j**) and *Etv5* (**k**) in mock or Satb2 expressing B progenitors. **l–n** Q-RT PCR analyses of expression of B-cell differentiation genes *Pax5* (**l**), *Ebf1* (**m**) and *Cdkn2a* (**n**) in mock or Satb2 expressing B-cell progenitors. **o** Chromatin immunoprecipitation using α-H3K27me3 antibody and Q-PCR analyses of the enrichment of *Pax5, Ebf1,* and *Cdkn2a* genes in mock/Satb2 transduced WT/DKO leukemic B-cell progenitors. **p** Chromatin immunoprecipitation using α-H3K4me3 Ab and Q-PCR analyses of the enrichment of *Pax5, Ebf1,* and *Cdkn2a* genes in mock/Satb2 transduced WT/DKO leukemic B-cell progenitors Data are presented as mean ± SD of a minimum of three independent experiments.*$p < 0.05$; **$p < 0.01$; ***$p < 0.001$, *t*-test

aPKC deficient B-cell progenitors (Supplementary Figure 10D). We observed a lack of reversibility of the expression of the pro-apoptotic *Bax* or *Bak* in leukemic B-cell progenitors (Supplementary Figure 10E), nuclear translocation of Numb

(Supplementary Figures 10F–G), and downregulation of Myc (Supplementary Figures 10H–I), indicating that these activities are Satb2 independent. Altogether, these data indicate that Satb2 is downstream of aPKC and Etv5, and that ectopic expression of

Satb2 is sufficient to rescue the proliferation and differentiation arrest, but not the survival, of leukemic B-cell progenitors in vitro and in vivo.

**aPKCι is required for human leukemic cell activity**. Our results, in an inducible mice model, demonstrate the indispensability of aPKCλ in the development and progression of P210-BCR-ABL+CML/B-ALL. We observed increased expression and activation of aPKCι in LSC/P from CML and B-ALL patients (Fig. 1). To investigate the roles of aPKCι in human leukemia, CD34+ cells from CML patients were transduced with aPKCι shRNA lentiviral vector containing EGFP as a reporter (Fig. 6a). Transduction of CML CD34+ cells with aPKCι shRNAs led to a 75% decrease in the expression of aPKCι (Fig. 6b). Depletion of aPKCι led to a 50% decrease in CFU formation of CML CD34+ cells (Fig. 6c). The absolute expansion in culture medium containing h-SCF, h-IL3, and h-GM-CSF was significantly inhibited (Fig. 6d). Downregulation of aPKCι led to increased differentiation and apoptosis of CML CD34+ cells (Fig. 6e, f). To analyze the role of aPKCι in human LSC/P survival and maintenance in vivo, CML CD34+ cells were transduced with control or aPKCι shRNA lentiviral vector followed by transplantation into sublethally irradiated NSG-SGM3 humanized mice (Fig. 6a). Mice transplanted with aPKCι shRNA transduced CML CD34+ cells showed a significantly reduced level of human chimera (hCD45+ EGFP+) in the BM (Fig. 6g). Further, the depletion of aPKCι led to reduced proliferation (Fig. 6h), increased apoptosis (Fig. 6i), and increased myeloid differentiation (Fig. 6j) of CML cells.

To determine the effect of the pharmacological inhibition of PKC in vivo, NSG-SGM3 mice were transplanted with CML CD34+ cells, and 10 days post-transplant, were treated with either the PKC inhibitor Ro-31-8200 or imatinib or both imatinib and Ro-31-8200, or treated with PBS (Control) (Fig. 6k). As shown in Fig. 6l, mice treated with either imatinib or Ro-31-8200 showed a significantly decreased levels of total human chimera (hCD45+) in the BM compared with PBS treated mice. However, the combination of both imatinib and Ro-31-8200 did not reduce the human chimera level further, indicating that imatinib and Ro-31-8200 may act on different signaling pathways. To further determine the effect of PKC inhibition on the survival and maintenance of an imatinib-resistant p210-BCR-ABL mutant[45], human cord blood CD34+ cells were transduced with T315I-p210-BCR-ABL retroviruses, and transplanted into NSG-SGM3 mice. Three weeks post-transplantation, mice were treated with imatinib or Ro-31-8200 or PBS. As expected, we observed similar levels of hCD45+ EGFP+ chimera in the imatinib- and PBS treated mice. However, mice treated with Ro-31-8200 showed significantly decreased levels of hCD45+EGFP+ chimera in the BM (Fig. 6m). These data demonstrate that aPKCι is essential for the maintenance of p210-BCR-ABL-induced leukemia and could be a novel therapeutic target for WT as well as imatinib-resistant leukemia.

## Discussion
The expression of oncogenic drivers in HSC/P results in transformation and generation of the so-called LSC/P, which give rise to B-lymphoid and myeloid malignancies. The most prominent defects in lymphoid leukemia genomes affect lineage-specific genes, as demonstrated in BCR-ABL+ B-cell precursor leukemia, where lymphoid lineage gene mutations frequently guide lymphoid proliferation and differentiation arrest[9–11]. BCR-ABL+ leukemias have provided and continue to provide information on the concepts of oncogenic and non-oncogenic addiction, and also have allowed the identification of a number of targetable mechanisms of resistance to existing targeted therapies[41,46].

Searching for the identification of basic mechanisms that control leukemic progenitor fate, we focused on the role of atypical members of the PKC family, which are involved in many aspects of fate specification and differentiation[12,47]. In particular, we examined their role in preB ALL development based on the circumscribed expression of BCR-ABL in HSC/P. While there is a good understanding of the role of classical and novel PKC isoforms[48–50], the role of aPKCs in the development and progression of leukemia is largely unknown. aPKCι has been shown to protect the BCR-ABL+ erythroleukemia cell line K562 against drug-induced apoptosis through the activation of NFkB/RelA pathway[15,16,51]. However, an evaluation of the distinct or redundant functions of aPKCζ and aPKCλ/ι in the development and progression of leukemia from the chronic to the blast crisis phases of the disease has remained unexplored due to the lack of appropriate animal models that recapitulate the human disease evolution.

This report identifies aPKCλ/ι as a crucial regulator of leukemogenesis, and indicates that the transformation and maintenance of lymphoid leukemogenesis depends on aPKCλ/ι and newly identified downstream signaling targets. We have shown that leukemic, but not normal, B-cell progenitor differentiation arrest is dependent on upregulated aPKC activity to which these cells become addicted.

Our study shows that aPKCλ/ι is predominantly expressed and activated in human CML and B-ALL progenitors as well as in BCR-ABL+ murine LSC/P. Activation of aPKCλ/ι depends on BCR-ABL TK and Cdc42 activities. The genetic deficiency of aPKCλ or both aPKCζ and -λ, but not aPKCζ alone, or the pharmacological inhibition of aPKCs, attenuate the development of a p210-BCR-ABL- or TKI-resistant p210-BCR-ABL-induced human and murine CML-like disease, and completely abrogate the progression to B-ALL. The expression of aPKCζ in leukemic B-cell progenitors is very low but functional, and follows the classical role of oncogenic suppression assigned to aPKCs[20–23,25].

Abrogation of B-ALL in the absence of aPKCλ is associated with impaired MEK/ERK signaling[52], and polarization and/or activation of fate determinants Numb and Lgl1[53,54]. Interestingly, Rac activity, a major regulator of leukemic signaling in transformed myeloid and lymphoid progenitors[29,55,56], does depend on aPKCλ, suggesting that aPKCλ/ι mediates the activation of a Rac targeting guanine nucleotide exchange factor. Further, our results show antagonistic roles for aPKCλ and aPKCζ in Myc protein regulation. While aPKCζ is a well-known negative regulator of Myc protein stability by its direct phosphorylation[33], the effect of aPKCλ/ι on Myc protein expression has not been well characterized. Our data, using combinations of genetic and pharmacological approaches, supports that aPKCλ/Rac/MEK-ERK are necessary for Myc activity while aPKCζ acts as a negative regulator of Myc.

Our data identify the axis aPKC/Satb2 mediated by ERK activation as a crucial regulator of leukemic B-cell differentiation arrest. It has been shown that some surviving BCR-ABL+ preB leukemic cell clones acquire permissiveness to oncogenic signaling by strong activation of the negative feedback regulation of ERK signaling. In studying the negative feedback regulation of Erk, the transcription factor Etv5 was found to be essential for oncogenic transformation in pre-B ALL[42]. We confirmed that Etv5 expression is absolutely dependent on MEK-ERK activity.

Our model recapitulates the human clonal evolution theory based on the selection of a "specific level" of Erk activation, since leukemic B-cell progenitors from secondary and tertiary recipient mice expressed a higher level of Etv5 than their normal counterparts, and these levels were downregulated to non-oncogenic levels upon deletion of aPKCλ or both aPKCλ and aPKCζ. Furthermore, Etv5 binds the Satb2 promoter in

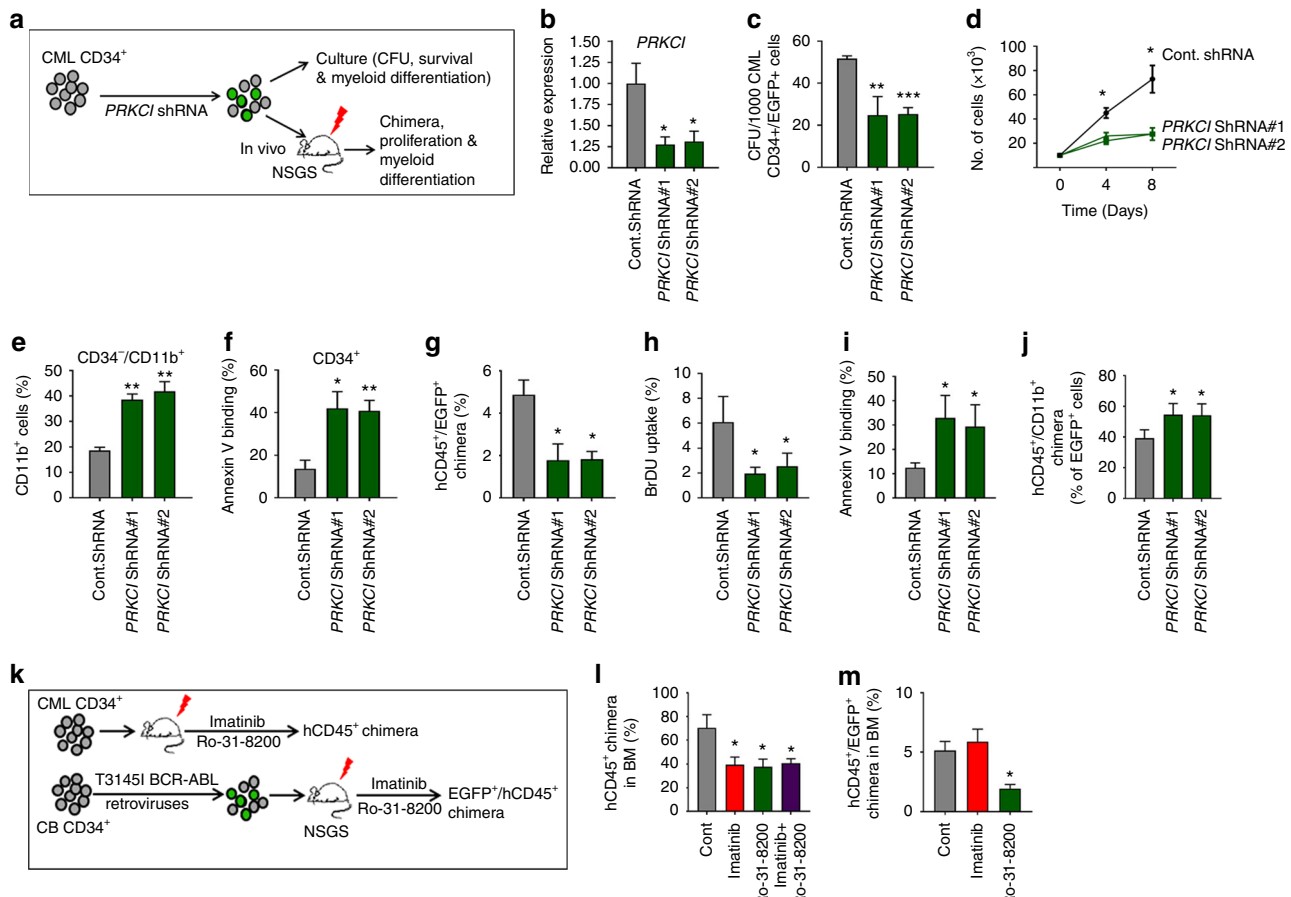

**Fig. 6** aPKCι is required for the survival and maintenance of human CML CD34+ cells. **a** Schematic diagram of shRNA mediated knockdown of aPKCι in CML patient-derived CD34+ cells followed by in vitro culture as well as in vivo transplantation into NSG-S mice. **b** Q-RT-PCR showing shRNA mediated knockdown of aPKCι in CML CD34+ cells. **c** Colony forming unit (CFU) cell assay showing reduced clonogenic efficiency of aPKCι deficient CML CD34+ cells. **d** Liquid culture expansion of control and aPKCι deficient CML CD34+ cells. **e** FACS quantification of CD34−/CD11b+ cells derived from control and aPKCι knocked down CML CD34+ cells grown in liquid culture medium. **f** FACS quantification of annexin V-binding of control/aPKCι knocked down CML CD34+ cells growing in liquid culture expansion medium. **g** hCD45+/EGFP+ chimera in NSG-S mice transplanted with CML CD34+ cells transduced with control/aPKCι shRNA lentiviruses. **h, i** Quantification of BrDU uptake (**h**) and annexin V-binding (**i**) of hCD45+/EGFP+ chimera in the bone marrow (BM) of NSG-S mice transplanted with control/aPKCι shRNA transduced CML CD34+ cells. **j** hCD45+/hCD11b+ on EGFP+ chimera in NSG-S mice transplanted with CML CD34+ cells transduced with control/aPKCι shRNA lentiviruses. **k** Schematic diagram of pharmacological administration of imatinib and a pan-PKC inhibitor (Ro-31-8200) to NSG-S mice transplanted with CML CD34+ cells and T315I p210 BCR-ABL transduced CB CD34+ cells. **l** FACS quantification of human CD45+ cell engraftment in the BM of NSG-S mice transplanted with CML CD34+ cells and treated with either control vehicle, imatinib, Ro-31-8200 or a combination of imatinib and Ro-31-8200. Imatinib and Ro-31-8200 administration led to a similar decrease of human CML burden in the BM of NSG-S mice. **m** FACS quantification of hCD45+/EGFP+ chimera in the BM of NSG-S mice transplanted with T315I transduced CB CD34+ cells and treated with control vehicle, imatinib or R0-31-8200. Mice treated with the PKC inhibitor show reduced levels of human CD45+/EGFP+ chimera in the BM, whereas imatinib treatment has no effect. Cont: Control. Data are presented as mean ± SD of a minimum of two independent experiments.*p < 0.05; **p < 0.01; ***p < 0.001, t-test

leukemic B-cell progenitors, and, when Etv5 is expressed in aPKC-deficient leukemic B-cell progenitors, it rescues Satb2 expression, leukemic proliferation, and B cell differentiation arrest. Forced expression of Satb2 is required for proliferation and differentiation arrest, but not survival, of leukemic B-cell progenitors in vitro and in vivo (Fig. 7). Re-expression of Satb2 rescues the epigenetic repressive activity of BCR-ABL on aPKC-deficient B-cell progenitor transcriptional hallmarks of differentiation, namely *Pax5*, *Ebf1* and *Cdkn2a*, as well as leukemogenesis. Our data identify aPKC as a major signaling factor controlling the expression of ERK-dependent epigenetic regulators (Etv5 and Satb2) of pre-B cell proliferation and B cell differentiation arrest, contributing to gene silencing programs with the ability to target B-cell lineage-specific genes.

We also demonstrate that Cdc42 activity is upregulated in BCR-ABL+ B-cell progenitors and its inhibition by casin abrogates aPKC activation. Rac is a mediator of Raf-independent MEK activation[52] and our data indicates that the Rac inhibitor NSC23766 is able to inhibit MEK/ERK activation of leukemic B-cell progenitors. This data is supported by the fact that *Etv5* expression is downregulated upon Mek, but not Raf, inhibition.

We demonstrate that aPKCλ/ι is required for BCR-ABL+ leukemia through its activity in regulating a signaling pathway that diverges from the classical view of dependence on Numb and Myc[13,54]. The fact that the deficiencies of aPKCλ/ι and/or aPKCζ do not modify the levels of self-renewal transcriptional signatures associated with BCR-ABL leukemogenesis[7,57,58], and the lack of effect of such deficiencies on normal B-cell differentiation in vivo, strongly suggest that the mechanism utilized by leukemic B-cell

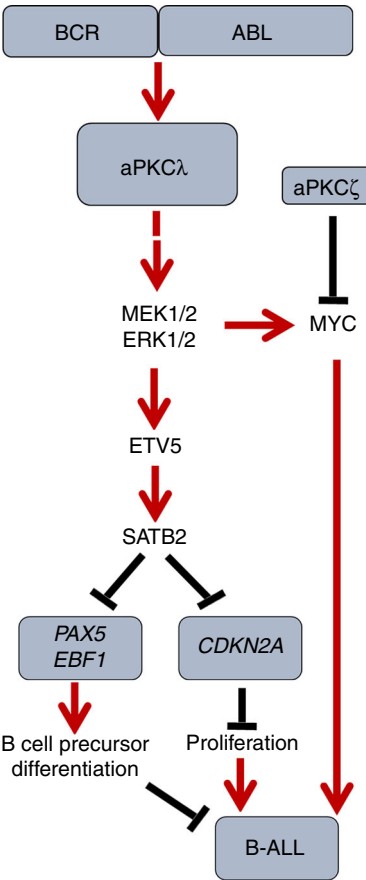

**Fig. 7** Schematic diagram of the roles of aPKCλ and –ζ in regulating B-ALL. The aPKCλ dependent Mek/Erk-Etv5-Satb2 axis plays a crucial role in the proliferation and differentiation of leukemic B-cell progenitors through Satb2 mediated epigenetic repression

progenitors differentiation arrest is distinct from the one reported in Numb/Myc-dependent fate determination in HSC.

The traditional view is that deletions of crucial genes that control proliferation, survival, and B cell differentiation are required for BCR-ABL+ B-ALL transformation, making the leukemic cells independent of BCR-ABL activity and therefore insensitive to TKI therapies. This view is supported by the evidence that B-cell blastic transformation in BCR-ABL+ leukemias is associated with genetic alterations to the determinants of the proliferation and lineage differentiation arrest. Deletion of *IKZF1* (IKAROS), *CDKN2A*, *PAX5* and *EBF1* genes are common features of *BCR-ABL+* lymphoid leukemia (CML at its progression to lymphoid blast crisis) but are rarely present in CML in its chronic phase[9–11]. These mutations, albeit prevalent, are not present in all cases of pediatric BCR-ABL+ B-ALL[10,59] suggesting that B-ALL transformation depends on an intermediate leukemogenic process independent of gene deletions. This view is also challenged by the observation that poor-prognosis Ph-like B-ALL can be effectively targeted by kinase inhibitors against downstream signaling pathways which are activated in Ph-like leukemias[60,61]. In addition, recurrent epigenetic alterations have been identified across all the B-ALL subtypes, and genes frequently affected by structural abnormalities have been shown to be targets for aberrant DNA methylation, suggesting that certain epigenetic events are required for leukemic transformation[62]. Despite all these advances, the mechanisms that drive B-cell differentiation arrest in the context of oncogenic signaling remain unknown. For instance, it remains unclear whether defects in

lineage-specific genes arise from the selection of a random mutation, or whether these genes are more likely to become defective. Boulianne et al. have demonstrated that lineage-specific genes are prominent hotspots for DNA damage during leukemic transformation of B-cell progenitors[63]. Our results suggest that these damaged DNA hotspots could also be marked before gene deletion occurs, and strongly suggest the development of a transitory phase in B-cell precursor leukemogenesis that would entail epigenetic-mediated silencing of a transcriptional network of genes crucially regulating B-lymphoid differentiation. The serial transplantation of Scl-tTA/TRE-BCR-ABL transgenic cells results in the selection of transformed leukemic progenitors with high levels of expression of BCR-ABL, similar to that which occurs in human blastic transformation[2]. A high level of BCR-ABL expression, without the loss of p53 is associated with CML transformation into B-ALL in ~70% of cases[64]. Our data strongly suggests that downstream upregulation of ETV5/SATB2 represents a major epigenetic regulator in the regulatory network involved in B-cell differentiation arrest, encompassing at least *Pax5, Ebf1,* and *Cdkn2a,* and this upregulation may directly or indirectly affect the expression of *Ikzf1, Ikzf3,* and *Rag1,* all crucial regulators of B-cell differentiation that are silenced during B-cell transformation.

Because CML lymphoid crisis is frequently resistant to conventional therapies, it is critical to identify the pathways that drive this aggressiveness. The fact that an aPKC deficiency does not impair normal hematopoiesis, as demonstrated in this report and in our previous publication[27], emphasizes the uniqueness of aPKC dependent activity in B cell precursor transformation. Atypical PKC deficient leukemic B-cell progenitors have decreased survival, but are able to persist for a long time in circulation, and to maintain a status of latent leukemogenesis. Inhibition of aPKC predominantly results in the loss of B-cell differentiation arrest and proliferation, suggesting that drugs targeting the kinase activity of aPKC may be useful in the treatment of the ever-challenging oncogenic and non-oncogenic dependencies of BCR-ABL+ leukemias[65,66]. Confirming this expectation, our data demonstrates that xenografted leukemic cells expressing the mutation T315I are susceptible to the administration of the PKC inhibitor Ro-31-8200.

In summary, this report highlights the role of aPKC in B-cell precursor leukemogenesis and unveils the role of SATB2 -dependent transcriptional repression activity in the specific development of B-ALL, with profound mechanistic and translational implications, in both our understanding of the process of leukemogenesis and the identification of alternative targets for Ph+-, and probably Ph-like, leukemias.

## Methods

**Human specimens**. De-identified normal BM, G-CSF mobilized PB, Ph+ pre-B ALL BM and CML PB mononuclear cell specimens were obtained from the Healthy Donor and Leukemia Bank repositories of Cincinnati Children's Hospital Medical Center under Institutional Review Board-approved protocols and with donor informed consent. All specimens (normal, B-ALL and CML) were derived from young adult individuals (age range: 18–39) with both genders represented. Material from therapeutic leukaphereses from two CML patients was submitted to CD34 selection by CliniMACS (Miltenyi Biotec Inc, Auburn, CA). Post-selection purity was >99%. Interphase FISH showed that >98% of CD34+ cells carried the t (9,22) translocation, respectively. BM from the Ph+ pre-B ALL specimens were sorted for CD34+/CD19+ B-cell progenitors and used for the isolation of total RNA.

**Scl-tTA; TRE-p210-BCR-ABL; Mx1-Cre; aPKCζ−/−; aPKCλ^F/F mice**. The generation of Scl-tTA; TRE-p210-BCR-ABL (Scl/p210) and Mx1-Cre; aPKCζ−/−; aPKCλ^F/F mice models have been described earlier (Koschmieder et al., Blood 2005; Sengupta et al., PNAS 2012). In brief, aPKCλ^F/F mice were crossed with Mx1-Cre mice to get Mx1Cre; aPKCλ^F/F genotypes. The Mx1-Cre; aPKCλ^F/F mice were further bred with aPKCζ−/− mice to get the triple transgenic Mx1Cre; aPKCζ−/−; aPKCλ^F/F mice. FVB/N-backcrossed Scl/p210 mice (kindly provided by Dr Claudia

Huettner, Blood Center of Southeastern Wisconsin, Milwaukee) were crossed with Mx1-Cre; aPKCζ$^{−/−}$; aPKCλ$^{F/F}$ mice to generate penta-transgenic Scl-tTA; TRE-p210-BCR-ABL; Mx1-Cre; aPKCζ$^{−/−}$; aPKCλ$^{F/F}$ mice. Mice were genotyped as described previously (Koschmieder et al., Blood 2005; Sengupta et al., PNAS 2012). Littermate mice from the same breeding were used in all experiments. CD45.1$^+$ B6. SJLPtprc$^a$ Pep3b$^b$/BoyJ (mentioned as BoyJ) and CD45.2$^+$ C57BL/6 mice were obtained from in-house mouse core facility at Cincinnati Children's Hospital Medical Center and Jackson Laboratory, respectively. All mouse strains were maintained at an Association for Assessment and Accreditation of Laboratory Animal Care accredited, specific-pathogen-free animal facility at Cincinnati Children's Research Foundation, Cincinnati, under an Institutional Animal Care and Use Committee approved protocol. All mice were between 6 and 12 wk of age at the time of experimentation.

**Serial transplantation and development of CML and B-ALL**. Scl/p210; Mx1-Cre; aPKCζ$^{−/−}$; aPKCλ$^{F/F}$ and littermate mice were maintained with a diet containing doxycycline (6 mg/g of food; Bioserve Biotech). Pooled BM cells from a minimum of 3 mice in each group were transplanted into lethally irradiated CD45.1$^+$ BoyJ recipients. Each recipient mouse was transplanted with $10 \times 10^6$ BM cells. The transplanted chimeric mice were fed with doxycycline containing diet (Dox on), and 4 weeks post-transplantation, PB CD45.2$^+$ chimera was analyzed. After the confirmation of more than 95% CD45.2$^+$ chimera in the PB, doxycycline from the diet was removed (Dox off) to induce p210-BCR-ABL expression. Four weeks after the induction of p210-BCR-ABL expression, mice were treated intraperitoneally with five injections of 10 μg/body kg weight polyinosinic:polycytidylic acid (polyI:C) (Amersham) on every other day for the inducible deletion of aPKCλ. The genomic DNA from PB collected after 7 days of last polyI:C injection was isolated to evaluate the level of aPKCλ deletion. A multiplex PCR amplifying the WT allele, the aPKCλ floxed allele, and the aPKCλ gene-deleted allele was performed in conditions of linear amplification (28 cycles) using previously described primers (Sengupta et al, PNAS 2012). The mice were followed up to 6 months for the development and progression of CML like disease, and the sick, hunched, and moribund mice were sacrificed, and the BM, spleen, and PB were analyzed.

BM cells ($5 \times 10^6$ cells/recipient) from the chimeric mice with CML like disease were competitively transplanted, along with an equivalent amount of CD45.1$^+$ BM cells, into BoyJ secondary recipients. The secondary recipient mice were followed for leukemia development, and mice manifesting B-ALL like symptoms such as hind leg paralyses, head and neck tumor, failure of central nervous system, massive egress of leukemic B-cell-precursors in to PB were sacrificed, and further analyzed. Whole BM cells from the secondary recipient mice were transplanted ($5 \times 10^6$/ mouse) into lethally irradiated tertiary recipients and were followed for the development of B-ALL, and the sick and moribund mice were sacrificed and analyzed further.

**Viral transduction and transplantation**. The lentiviral vectors Satb2-IRES-EGFP (pReceiver-Lv165-Satb2) and Etv5-IRES-EGFP (pReceiver-Lv165-Etv5) for Satb2 and Etv5 expression, respectively, and the mock vector (pReceiver-Lv165) were purchased from Genecopoeia. The lentiviral vectors for aPKCι shRNA (pLKO.1-CMV-tGFP shRNA plasmid DNA, catalog SHCLND 10141531MN) and control shRNA (pLKO.1-puro-CMV-TurboGFP$^+$ control plasmid DNA, catalog SHC003 02101412MN) were obtained from Sigma-Aldrich. High titter lentiviruses of Satb2, Etv5 and mock vector were generated in 293 T cells. Lin$^−$ Sca1$^+$ cKit$^+$ (LSK) cells were isolated from WT and DKO primary leukemic mice, and were transduced with Satb2-IRES-EGFP (Satb2) or IRES EGFP (mock) lentiviruses at a multiplicity of infection (MOI) ratio of 25 in cultures containing recombinant mSCF (50 ng/mL), recombinant human TPO (25 ng/mL) and recombinant human Flt3L (25 ng/mL) (all from Peprotech Inc.). Five thousand transduced, unsorted cells were transplanted into lethally irradiated (7 + 4.75 Gy) C57Bl/6 secondary recipient mice. Recipient mice were followed for the development of B-ALL, and the sick and moribund mice were sacrificed and analyzed.

**Human xenograft model, and Imatinib and Ro-31-8220 treatment**. NOD-scid IL2Rgnull -3/GM/SF (NSG-SGM3 or NSG-S) mice were transplanted with CML patient-derived CD34$^+$ cells or human Cord blood-derived CD34$^+$ cells transduced with a T315I-p210-BCR-ABL mutant expressing retroviral vector. Ten days post-transplantation, Alzet osmotic pumps (Model 2002, Durect,Cupertino, CA) containing either PKC inhibitor Ro-31-8220 (1 mM) or imatinib (0.5 mM) or combination of both imatinib and R0-31-8220 or PBS were subcutaneously implanted into recipient mice at a flow rate of 0.5 ml/h for 14 days. At the end of the 14 days of the administration of imatinib or Ro-31-8200 or combination of both imatinib and Ro-31-8220, mice were sacrificed and FACS-analyzed for the presence of h CD45$^+$ chimera in case of CML CD34 + cells transplanted group and hCD45$^{+/}$EGFP$^+$ chimera in T315I-p210-BCR-ABL transduced CB CD34$^+$ cells transplanted mice. Imatinib (Catalog 1-5508) and Ro-31-8220 (Catalog R136) were purchased from LC Laboratories and Sigma-Aldrich, respectively.

For human xenograft model of chronic myelogenous leukemia, CML patient's derived CD34$^+$ cells were transduced with control shRNA/aPKCι shRNA lentiviruses, and then transplanted into sub lethally irradiated (350 Gy) NSG mice. Immunophenotypic characterization of human chimera in the transplanted mice

was assessed by flow cytometric analyses using antibodies (from BD biosciences) against hCD45 (clone HI30), hCD34 (clone 581), hCD11b (clone ICRF44), hCD19 (clone HIB19), hCD33 (clone HIM3-4).

The role of atypical PKC in the development of Imatinib-resistant p210-BCR-ABL$^+$ leukemia in vivo in a human xenograft mouse model was evaluated by administering imatinib (0.5 mM) or the PKC inhibitor Ro-31-8220 (1 mM) using Alzet osmotic pumps (Model 2002, Durect,Cupertino, CA) for constant and control release of the drugs. Humanized NSG-S mice were transplanted with human Cord blood-derived CD34 + cells transduced with a T315I-p210-BCR-ABL mutant retroviral vector. Ten days post-transplantation, Alzet osmotic pumps containing imatinib (0.5 mM), the PKC inhibitor Ro-31-8200 or PBS (vehicle) were subcutaneously implanted into recipient mice at a flow rate of 0.5 ml/h for 14 days. At the end of the 14 days of the administration of imatinib, Ro-31-8200 or vehicle PBS, mice were sacrificed and FACS-analyzed for hCD45$^{+/}$EGFP$^+$ chimera in T315I-p210-BCR-ABL transduced CB CD34$^+$ cells transplanted mice.

**In vitro differentiation of B-cell precursors**. The in vitro differentiation of LSC (CD45.2$^+$Lin$^−$c-Kit$^+$Sca1$^+$) towards B-cell precursors was carried out by co-culturing the LSC on OP9 mouse stromal cell monolayers. OP9 mouse stromal cells were maintained in IMDM medium containing 20% FBS, 2 mM Glutamine and $10^{−4}$M β-Me. The LSK cells derived from WT and DKO chimeric mice were transduced with Etv5-IRES-EGFP (Etv5) or IRES EGFP (mock) lentiviruses at 25 MOI. The transduced cells were co-cultured on OP9 stromal monolayer in IMDM cell culture medium containing 5% FBS (Pre B cells defined), 50 ng/mL SCF and 10 ng/mL IL-7 for 7 days. After 7 days culture in presence of SCF and IL-7, cells were maintained in IMDM medium containing 5% FBS and 10 ng/mL IL-7 for B-cell precursor's development. At various time points, a fraction of cells from the culture was FACS-analyzed for immunophenotypic appearance of B-cell precursors (CD45$^+$B220$^{lo}$CD43$^+$CD19$^+$).

**Immunophenotypic analysis, sorting, and phospho-FACS analyses**. To label hematopoietic stem and progenitors, PB, BM, spleen cells were stained using a cocktail of biotin-conjugated anti-mouse lineage mAbs CD45R (B220, Clone RA3-6B2), Gr-1 (Ly6G, Clone RB6-8C5), CD4 (L3T4, Clone GK1.5), CD8a (Ly-2, Clone 53-6.7), Mac-1 (CD11b, CloneM1/70), CD3_ (Clone 145- 2C11), and TER119 (Ly-76), all from BD Pharmingen. Cells were washed and then stained with APC-Cy7-streptavidin, PECy7–anti–mouse Sca-1 (Ly6A/E, clone D7), APC–anti-mouse CD117 (c-kit, Clone 2B8), Pacific Blue–anti-mouse CD34 (RAM 3-4) and PE–anti-mouse Flk2 antibodies, all from BD Pharmingen.

For lineage distribution analyses the PB, BM, and splenocytes of primary, secondary and tertiary recipient leukemic mice were stained using FITC-anti-mouse-CD45.2 (clone 104), APC-anti-mouse CD11b (clone M1/70), PE-Cy7-anti-mouse CD3e (clone 145-2C11), APC-Cy7–anti–mouse CD45R (clone RA3-6B2), and then flow analyzed. For the analyses of B-precursors, PB, BM and splenocytes of primary, secondary and tertiary recipient leukemic mice were stained using FITC-anti-mouse-CD45.2 (clone 104), APC-Cy7–anti-mouse CD45R (clone RA3-6B2), PerCP-Cy5.5–anti mouse IgM (clone II/41), PE–anti-mouse CD43 (clone S7), PE-Cy7- anti-mouse CD19 antibodies. For the immunophenotypic characterization of B-precursor cells in mock/Satb2 transduced and transplanted mice, the PB, BM, and splenocytes were stained for APC-anti-mouse-CD45.2 (clone 104), APC-Cy7–anti-mouse CD45R (clone RA3-6B2), PerCP-Cy5.5–anti-mouse IgM (clone II/41), PE–anti-mouse CD43 (clone S7), PE-Cy7- anti-mouse CD19 antibodies, and EGFP$^+$ B-precursors were analyzed. Fluorochrome tagged antibodies for flow cytometry analyses were obtained from BD Biosciences. FACS-Canto flow cytometer and the FACSArial cell sorter (both BD Biosciences) were used for analyses and sorting, respectively.

For phospho-FACS analyses of activated aPKC (phopsho-aPKC), cord blood-derived CD34$^+$ cells and B-ALL patients' derived BM cells were stained for surface markers (hCD45/hCD34/hCD19) followed by fixation of with cytofix solution (BD Biosciences) for 30 mic on ice. The fixed cells were permeabilized using cytofix/cytoperm permeabilization reagent and then washed twice with 1X Perm/wash solution containing 0.5% BSA. Cells were treated with isotype control IgG (Cell Signaling technologies, catalog 3900) or phospho-aPKC-T410/403 (Cell Signaling technologies, catalog 9378) primary antibodies. Cells were washed and treated with Alexa-647 conjugated anti-rabbit IgG for 30 min on ice. Cells were washed, resuspend with PBS and FACS analyzed. The ratio of the mean fluorescence intensity of Alexa fluor-647 between phospho-aPKC and control IgG-treated hCD34$^+$ hCD19$^+$ B progenitors was evaluated and presented. The gating strategy of FACS analyses and sorting of murine and human stem and progenitors are depicted in supplementary information.

**Confocal immunofluorescence microscopy**. The sorted B cell progenitors derived from Scl/p210; WT/aPKCζ$^{−/−}$/aPKCλ$^{Δ/Δ}$DKO chimeric mice and mock/Satb2 transduced/transplanted mice were seeded onto fibronectin (RetroNectin catalog T100B, TAKARA BIO INC.) coated glass chamber slide in culture medium containing mouse stem cell factor (50 ng/mL) and mouse-IL7 (10 ng/mL). Cells were allowed to adhere to the fibronectin-coated glass chamber slide for 4 h, and then fixed using 4% paraformaldehyde for 30 min at 4°C, permeabilized with 0.1% Triton X-100 (catalog T9284, Sigma-Aldrich) for 10 min followed by blocking with 5% protease-free bovine serum albumin in PBS for 30 min. The slides were stained

with primary antibodies; anti-Numb (catalog ab4147, dilution- 1:500), anti-phospho-Llgl1 (catalog ab59950, dilution-1:500) or anti-Myc (Cell Signaling Technologies, Catalog 13987, dilution-1:250) at 4 °C overnight. The cells were washed and then treated with secondary antibodies (from Life technologies) donkey anti-rabbit Alexa Fluor 488 (catalog A11034, dilution-1:500) or donkey anti–goat Alexa Fluor 568 (catalog A11041, dilution-1:500) at 1:500 v/v concentration for 1 h at room temperature. Cells were washed and mounted using Gold Antifade mounting media (catalog P36935, Life technologies) containing DAPI. The stained cells were analyzed by a LSM 710 confocal microscope system (Carl Zeiss) equipped with an inverted microscope (Observer Z1, Zeiss) using a Plan Apochromat × 63 1.4 NA oil immersion lens. The images were processed using Adobe Photoshop v7.

**Proliferation and survival assays.** Proliferation and Survival of B-precursor cells were determined by in vivo BrDU uptake and annexin V binding assay, respectively. For in vivo proliferation analysis, mice received single intraperitoneal injections of BrdU (500 µg). Forty-five minutes later, BM cells were harvested and stained for surface markers (B220-APC-Cy7; CD19-PE-Cy7; CD43-PE; IgM⁻PerCp-Cy-5.5; CD45.2-FITC). The cells were then fixed, permeabilized and stained with allophycocyanin-conjugated anti- BrdU antibody according to the manufacturer's instructions (BD Biosciences) and then FACS analyzed. To analyze the proliferation of human CML cells transplanted into NSG-S humanized mice, at day 21 post-transplantation xenograft mice were injected intraperitoneally with BrDU (1 mg/20 g mice) for 45 min and BM cells were stained for cell surface markers, and then processed for BrDU staining followed by FACS analyses. The annexin V binding was performed as described in the manufacturer's protocol (BD Biosciences)

**CFU-proB assays.** B-cell lineage colony-forming units (CFU-proB) efficiency of BM cells, splenocytes and PB were evaluated using standard condition. In brief, BM cells, splenocytes, RBC lysed PB or sorted B-precursors were cultured in methylcellulose semisolid medium (catalog M3134,StemCell Technologies) supplemented with 30% FBS (for mouse B lymphoid colony forming cells; Stem Cell Technologies), 2mM L-glutamine (Invitrogen), 1% antibiotics (penicillin-streptomycin; Invitrogen), $10^{-4}$ M β-mercaptoethanol (Fisher-Scientific), 1% BSA (Sigma-Aldrich), 20 ng/mL of recombinant mouse IL-7 (PeproTech), and 100 ng/mL of recombinant mouse SCF (PeproTech), and CFU-proB quantification at day9 was evaluated.

**Western blot analyses.** Normal B cell progenitor (B220lo IgM⁻CD43⁺CD19⁺) cells were sorted from LDBM cells of WT/aPKCζ⁻/⁻/aPKCλΔ/Δ/ DKO mice, and leukemic B-cell progenitors (CD45.2⁺ FSChi B220lo IgM⁻ CD43⁺ CD19⁺) were sorted from LDBM cells of chimeric mice belonging to Scl/p210; WT/ aPKCζ⁻/⁻/aPKCλΔ/Δ/ DKO group. CML patient-derived stem and progenitor cells (CML CD34⁺ cells) from two different CML patients (specimens were obtained through institutional review board-approved protocols from Hoxworth Blood center, Cincinnati) and G-CSF mobilized healthy donor peripheral blood (PB) CD34⁺ cells (obtained from Translational Trials Development and Support Laboratory of Cincinnati children's hospital medical center) were taken. Whole cells lysates of above-mentioned cells were prepared by lysing sorted cells in 1X RIPA buffer (Cell Signaling Technologies, Catalog 9806) containing protease inhibitor cocktail (Roche, Catalog 04693159001) and phosphatase inhibitors (Roche, Catalog 046906837001). Equal quantity of protein from each samples were dissolved in Laemmli buffer, boiled for denaturation, and then samples were electrophoresed through 4%–15% SDS-PAGE gradient gel followed by transfer to PVDF membrane. For sensitive detection of aPKCζ, higher amounts of protein (20 µg whole cell lysate in case of CD34+ cells, and 50 µg in case of murine B-cell progenitors) were used. The membranes were blocked and treated with primary antibodies against aPKCζ (Catalog 9368), phospho-aPKCζ/λ (Thr410/403; Catalog 9378), c-Abl (Catalog 2862), phospho-Erk1/2 (Thr202/Tyr204, Catalog 4377), Erk1/2 (Catalog 9102), phospho JNK (Thr183/Tyr185, Catalog 9251), JNK (Catalog 9252), Mek1/2 (Catalog 9122), phospho Mek1/2 (Ser221, Catalog 2338), phospho-P38 (Thr180/Tyr182, Catalog 9216), P38 (Catalog 9212), phospho-b-Raf (Ser445, Catalog 2696), phospho-cRaf (Ser289/296/301, Catalog 9431), phospho-p65 (Ser536, Catalog 3031), p65 (Catalog 6956), phospho Numb (Ser276, Catalog 9878), Numb (Catalog 2756), c-Myc (Catalog 13987) from Cell Signalling Technologies at 1:1000 dilution, and aPKCλ (Catalog 610208, dilution-1:500) from BD Transduction Laboratories, β-actin (catalog A5441, dilution-1:5000) from Sigma-Aldrich, followed by washing and subsequent treatment with secondary antibodies tagged with HRP [anti–mouse IgG, catalog 70765, Cell Signaling Technologies; anti-rabbit IgG, catalog 70745, Cell Signaling Technologies at 1:1000 dilution). The blots were developed using a chemiluminescence coupled reaction. The band intensities on the X-ray films were quantitated by using ImageJ software, and normalized against actin band intensity of the corresponding sample.

For the pharmacological inhibition of Cdc42 and Rac GTPases, WT leukemic B-cell progenitors were treated with casin (10 µM) and NSC23766 (50 µM), respectively, for different time periods. For the Raf and Mek inhibition, WT leukemic B-cell progenitors were treated with 10 nM debrafenib (Selleckchem.com, Catalog S2807) and 10 nM trametinib (Selleckchem.com, Catalog S2673) for

different time periods. To analyze role of Mek-Erk activation on Myc protein expression, WT, aPKCζ⁻/⁻, DKO leukemic B-cell progenitors were treated with trametinib for different time periods. For the inhibition of BCR-ABL activity, cells were treated with 1 mM imatinib. At each time point, cells were washed with ice-cold PBS and lysed in 1X RIPA buffer containing protease inhibitor cocktail and phosphatase inhibitors on ice for 15 min. Equal quantities of protein from each sample were dissolved in Laemmli's buffer, boiled for denaturation, and then samples were electrophoresed through 4%–15% SDS-PAGE gradient gel followed by transfer to PVDF membrane. The membranes were processed for western blot analyses as described above. The images of whole western blots are presented in Supplementary Information.

**Rac and Ras activation assays.** B cell progenitors derived from leukemic mice were processed for the evaluation of activated Rac and Ras GTPases. Cells were pelleted and flash-frozen, and the proteins were isolated with 1X Magnesium Lysis/ Wash Buffer (Upstate Biotechnology, Lake Placid, NY) supplemented with 25 mM NaF and 1 mM Na3VO4. GTP-bound Rac from the lysates was immunoprecipitated using PBD (Pak1 binding domain) agarose beads (Upstate Biotechnology) and RBD (Raf1 Ras binding domain) agarose beads, according to the manufacturer's instructions. The immunoprecipitates were separated by sodium dodecyl sulfate (SDS)-polyacrylamide gel electrophoresis (PAGE) on a 12% gel (Bio-Rad, Hercules,CA) and were transferred to PVDF membrane followed by western blot analyses using antibodies against Rac (BD transduction laboratories, Catalog 610650) and Ras GTPases (BD transduction laboratories, Catalog 610001). Membranes were incubated with primary antibodies overnight at 4 °C, followed by incubation with a secondary antibody conjugated to horseradish peroxidase (HRP) and directed against mouse, rabbit IgG (Cell Signaling technologies) for 1 h at room temperature. The blots were developed using a chemilumiscence coupled reaction. The band intensities on the X-ray films were quantitated by using ImageJ software, and normalized.

**Transcriptome and bioinformatics analysis.** Total RNA was extracted from sorted leukemic B-cell progenitors derived from WT/aPKCζ⁻/⁻/aPKCλΔ/Δ/DKO leukemic chimeric mice using RNeasy Mini Kit (QIAGEN). RNA quality and concentration were measured by Bioanalyzer 2100 using the RNA 6000 Nano Assay (Agilent Technologies). RNA-seq libraries were prepared using the Illumina TruSeq RNA preparation kit and sequenced on the Illumina HiSeq 2000 using single-end 20 million–bp reads (Illumina). Reads were aligned with TopHat software, using hg19 as the reference genome and mapping reads per kilobase per million mapped reads (RPKM) as output. RPKM were log2-transformed and baselined to the median expression of the average of each class of samples. RNA samples were processed for RNAseq analyses using RNA-seq protocols from NuGEN and Illumina. The amplified products were sequenced to analyze the gene expression profile. The transcriptome data were further analyzed for differential expression and gene ontology (GO) pathway analyses using Strand NGS bioinformatics analyses software and Altanalyze software (developed by Cincinnati Children's Research Foundation).

**Quantitative RT-PCR.** Total RNA was extracted from normal B-cell progenitors derived from Mx1Cre; WT/aPKCζ⁻/⁻/aPKCλΔ/Δ/DKO mice, leukemic B-cell precursors derived from Scl/p210; Mx1Cre; WT/aPKCζ⁻/⁻/aPKCλΔ/Δ/DKO chimeric mice, leukemic B-cell progenitors derived from mock/Satb2 transduced and transplanted mice, debrafenib or imatinib treated leukemic B-cell progenitors, healthy donor mobilized PB derived CD34⁺ cells, CML patient-derived CD34⁺ cells and B-cell acute lymphoblastic leukemia (B-ALL) patient-derived B-cell progenitors (hCD45⁺/hCD34⁺/hCD19⁺), using RNeasy minikit (QIAGEN). The cDNA was prepared using Taq Man reverse transcription reagent (Applied Biosystems, Life Technologies, catalog N8080234). The mRNA expression levels of *Prkcz (PRKCZ), Prkci (PRKCI), Pax5, Ebf1, Ikzf1, Ikzf2, Ikzf3, Rag1, Satb1, Satb2 (SATB2), Cdkn1a, Cdkn1b, Cdkn2a, Ccnd1*, were analyzed by Q-RT-PCR assay using TaqMan Universal PCR master mix and TaqMan primers for specific genes (Roche Applied Science, Life technologies). The expression level was normalized with the expression of internal control gene Gapdh (GAPDH).

**Chromatin immunoprecipitation (ChIP) and ChiP Q-PCR.** B-cell progenitor cells (B220loCD19⁺Cd43⁺IgM⁻) sorted from the BM of leukemic chimeric mice, were processed for chromatin immunoprecipitation using antibodies against Satb2 (Abcam, catalog ab34735), Etv5 (Abcam, ab 102010), as described in manufacturer's protocol (Magna ChIP A/G, Millipore). Briefly, $2 \times 10^6$ cells for each samples were crosslinked using 1% paraformaldehyde (PFA), treated with glycine to neutralize excess PFA, and then cells were washed with ice-cold PBS followed by lyses to prepare nuclear fraction. The nuclear fraction was lysed in lyses buffer containing protease inhibitor cocktail, and homogenized using a dounce homogenizer. The nuclear lysates containing whole chromatin were transferred into a specialized Eppendorf (Diagenode), and then sonicated by using a Diagenode sonicator by placing samples on ice water bath. Shearing of the chromatin was checked by running a fraction of the sample on 0.8% agarose gel. The sonicated samples were diluted using ChIP dilution buffer, and 1% of the diluted chromatin from each sample was saved to use as input. Specific antibody or isotype control

IgG were added into the samples, incubated for 3 h followed by addition of ProteinA/G magnetic beads into each tube. After 3 h incubation, chromatin immune complexes bound to protein A/G magnetic beads were washed sequentially using low salt buffer, high salt buffer, LiCl wash buffer, and TE buffer. After last wash, samples along with the input saved (1%) before adding antibody to the sample were resuspended in 100 μl ChiP elution buffer containing proteinase K, and then incubated at 62 °C for 2 h followed by 10 min incubation at 95 °C to reverse cross-link and separate the chromatin from magnetic beads. The fragmented chromatin are eluted using a spin elution column provided with the manufacturer's kit. The ChiP eluted DNA was used as template to quantify the enrichment of the promoter region of genes *Pax5, Ebf1, Cdkn2a* in the case of Satb2 ChiP, and enrichment of *Satb2* promoter region in case of Etv5 ChiP by running quantitative real time PCR using custom made Taqman primers. For ChiP-QPCR quantification of activation and repression marks in BCR-ABL+ leukemic B-cell progenitors derived from WT and DKO chimeric mice or mock/Satb2 transduced and transplanted mice, chromatin immunoprecipitation using antibodies against H3K27me3 (Millipore, catalog 07-449), H3K4me3 (Diagenode, catalog C15410035) were carried out, and the enrichment of the promoter regions of *Pax5, Ebf1*, and *Cdkn2a* genes was evaluated. The custom made Taqman Primers (Forward primer, reverse primer and probes) used for gene enrichment analyses were obtained from Integrated DNA Technologies (IDT). The sequences of primers used for gene enrichment analyses are; *Actb*: forward primer-5'-CACGCTAGGCGTAAAGTTGG-3', reverse primer-5'-ACTAGCCACGAGAGAGCGAA-3', probe- FAM/AGTGATCCTCAGGACCC; *Gapdh*: forward primer-5'-AGCTACGTGCACCCGTAAAG-3', reverse primer-5'-GTGCGTGCACATTTCAAAAA-3', probe-5'-FAM/CCCTCTCTTTGGACCCGC; *Pax5*: forward Primer- 5'-CTCTGGCTAGCTGAGTGTTAAG-3', reverse Primer-5'-GGGCCACTAATCCCAGTATTT-3', probe-FAM/ACCGATGTAACC-CATGCTAGGCAA; *Ebf1*:forward primer- 5'-GAGAGCCTGCAAAT-CACTTCTA-3', reverse primer-5'-AAGAGAGGGAGAGGGAGAAAGAG-3', probe-FAM/AACAGAATCTCCATCCAGCAGGGT; *Cdkn2a*: forward primer-5'-CTACGTGAA CGTTGCCCATC-3', reverse primer-5'-GGCTTCTTTCTTGGG TCCTG-3', probe-FAM/GAGGGA TCACGGCCAGA; *Satb2*: forward primer-5'-TACGATTCTGAACACAGTGAAA-3', reverse primer-5'-AGAAGAGGAAACA CTTTACCAT-3', probe-FAM/ACGGCAATGTTTCCTTAATATTGCC.

## Data availability

All sequencing data (raw and processed) that support the finding of this study are accessible through the National Center for Biotechnology information Gene expression Omnibus (GEO), GSE121018. All other relevant data are available from the corresponding author on request.

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

## Acknowledgements

The authors want to thank the Cincinnati Children's Hospital Medical Center and Hoxworth Blood Center for their continued support. The authors want to acknowledge their gratitude to Dr. Michael Leitges from the University of Oslo for sharing aPKCλ f/f transgenic mice for breeding. The authors also want to thank Ms. Margaret O'Leary for editing the manuscript as well as the technical assistance of Jeff Bailey and Victoria Summey from the Mouse Core, and the Flow Cytometry Core, both supported by the National Institutes of Health P30 DK090971. We thank Kashish Chetal for helping in the bioinformatics analyses. This study has been partly supported by the National Blood Foundation (R.C.N.), National Institutes of Health F31HL1324801 (M.J.A.), R01CA172025 (J.M.), R01CA207177 (J.M.), R01GM110628 (J.A.C.), United States Department of Defense (W81XWH-15-1-0344, D.R.), Leukemia and Lymphoma Society of North America (J.A.C.) and Williams Lawrence & Blanche Hughes Foundation (J.A.C.).

## Author contributions

R.C.N. conceived and performed experiments, analyzed the data, interpreted experiments and wrote the manuscript. S.H. performed experiments. M.J.A., A.M.W., and F.M. performed technical support to generate crucial tools for experiments. J.P. collected, characterized and provided primary human specimens. M.R.-C. performed bioinformatics analysis. D.R., Y.Z., M.T. D.-M., and J.M. provided crucial reagents and wrote the manuscript. J.A.C. conceived experiments, analyzed the data, interpreted experiments and wrote the manuscript. The authors declare no competing interests.

## Additional information

**Competing interests:** The authors declare no competing interests.

