## [Peer Review File · Nature Communications]

This manuscript has been previously reviewed at another journal that is not operating a transparent peer review scheme. This document only contains reviewer comments and rebuttal letters for versions considered at Nature Communications. Mentions of the other journal have been redacted.

Reviewers' Comments:

Reviewer #1:

Remarks to the Author:

This is now a properly edited/converted version describing an interesting study. The data largely supports the conclusions drawn although the interpretation of the added Raf inhibition data is not really consistent with the conclusions drawn. This is a relatively minor issue but I think does a disservice to the rest of the paper.

Reviewer #3:

Remarks to the Author:

The authors have addressed all my comments and concerns.

Reviewer #4:

Remarks to the Author:

In the current manuscript the authors present their findings examining the role of aPKC ϵ in controlling leukemic B-cell precursor differentiation and survival. They demonstrate that aPKC ϵ controls MEK/ERK regulating Etv5, which controls Satb2 expression whereas the Prkcz has tumor suppressive function. On the basis of this and their previously published work on distinction between physiologic and pathologic B cell precursor maintenance and differentiation the findings are novel and of interest finding and potentially provides a therapeutic target, which would be of interest in imatinib resistant disease.

I did not see the manuscript in its previous format, but it is clearly markedly improved in terms of presentation and in the additional data presented to address the reviewers comments. In particular, the additional experiments showing that Trametinib abrogates MEK/ERK activation and Etv5 expression, and that NSC23766 inhibits Rac provide support for their hypothesis.

The manuscript and rebuttal letter clearly outline how they have addressed each of the reviewers' comments. To my reading they have not addressed, or addressed only tangentially, the question of tumor initiation versus maintenance, but I accept their comment that this is less relevant for target validation and in fact agree that they have focused their work on an important question for which we have not previously had good models.

There remain a few minor grammatical and language usage issues which can readily be addressed.

They should provide the rationale why they focused on Satb1 and Satb2 in the text as they have in their letter.

Some of the discussion seems repetitive and could be shortened without affecting content.

It will be necessary to see the data deposited in GEO when accepted.